# Cocktails of NSAIDs and 17α Ethinylestradiol at Environmentally Relevant Doses in Drinking Water Alter Puberty Onset in Mice Intergenerationally

**DOI:** 10.3390/ijms24065890

**Published:** 2023-03-20

**Authors:** Pascal Philibert, Stéphanie Déjardin, Mélissa Girard, Quentin Durix, Anne-Alicia Gonzalez, Xavier Mialhe, Mathieu Tardat, Francis Poulat, Brigitte Boizet-Bonhoure

**Affiliations:** 1Développement et Pathologie de la Gonade, Institut de Génétique Humaine, Centre National de la Recherche Scientifique (CNRS), Université de Montpellier, 34090 Montpellier, France; 2Laboratoire de Biochimie et Biologie Moléculaire, Hôpital Carèmeau, CHU de Nîmes, 30900 Nîmes, France; 3IExplore-RAM, Institut de Génomique Fonctionnelle, Centre National de la Recherche Scientifique, Université de Montpellier and Institut National de la Santé Et de la Recherche Médicale (INSERM), 34090 Montpellier, France; 4MGX-Montpellier GenomiX, UMS Biocampus, Université de Montpellier, CNRS, INSERM, 34090 Montpellier, France; 5Biologie des Séquences Répétées, Institut de Génétique Humaine, Centre National de la Recherche Scientifique, Université de Montpellier, 34090 Montpellier, France

**Keywords:** NSAIDs, EE2, environmental doses, puberty, testis, ovary, mice, gonadal transcriptome

## Abstract

Non-steroidal anti-inflammatory drugs (NSAIDs) and 17α-ethinyl-estradiol (EE2) are among the most relevant endocrine-disrupting pharmaceuticals found in the environment, particularly in surface and drinking water due to their incomplete removal via wastewater treatment plants. Exposure of pregnant mice to NSAID therapeutic doses during the sex determination period has a negative impact on gonadal development and fertility in adults; however, the effects of their chronic exposure at lower doses are unknown. In this study, we investigated the impact of chronic exposure to a mixture containing ibuprofen, 2hydroxy-ibuprofen, diclofenac, and EE2 at two environmentally relevant doses (added to the drinking water from fetal life until puberty) on the reproductive tract in F1 exposed mice and their F2 offspring. In F1 animals, exposure delayed male puberty and accelerated female puberty. In post-pubertal F1 testes and ovaries, differentiation/maturation of the different gonad cell types was altered, and some of these modifications were observed also in the non-exposed F2 generation. Transcriptomic analysis of post-pubertal testes and ovaries of F1 (exposed) and F2 animals revealed significant changes in gene expression profiles and enriched pathways, particularly the inflammasome, metabolism and extracellular matrix pathways, compared with controls (non-exposed). This suggested that exposure to these drug cocktails has an intergenerational impact. The identified Adverse Outcome Pathway (AOP) networks for NSAIDs and EE2, at doses that are relevant to everyday human exposure, will improve the AOP network of the human reproductive system development concerning endocrine disruptor chemicals. It may serve to identify other putative endocrine disruptors for mammalian species based on the expression of biomarkers.

## 1. Introduction

Pharmaceutical use has increased in developed countries over the last decades, both for human and veterinary therapy [1,2]. Non-steroidal anti-inflammatory drugs (NSAIDs) are over-the-counter molecules commonly prescribed to treat inflammation and pain [3]. Moreover, synthetic hormones are widely used to treat acne, endometriosis, and ovarian cysts, and are also used in combination with progestins for contraception and for hormone replacement therapy [4]. NSAIDs exert their pharmacological effects by inhibiting one or both cyclooxygenase isoforms that are involved in the conversion of arachidonic acid for prostaglandin synthesis [5]. Like natural hormones, synthetic estrogens display estrogenic activity by binding to estrogen receptors a (ESR1) and b (ESR2). They might be more resistant to metabolism and inactivation than the natural hormone 17β-estradiol (E2) (E2) [4,6]. These molecules can cross the placental barrier [7] and are metabolized in the liver by cytochrome P450-like enzymes, most often producing inactive metabolites [8,9]. Finally, these drug residues not used or detoxicated by the hepatic metabolism are excreted through the urine and faeces and are released in wastewater discharges. The efficiency of their removal in wastewater treatment plants varies depending on the molecules [10,11], leading to their release in the natural environment, particularly in surface [12,13] and drinking water (DW) [14,15].

Due to their huge and increasing use, NSAIDs such as ibuprofen (IBU, 2-(4-Isobutylphenyl) propionic acid)) and diclofenac (DCF, o-(2,6-Dicloroanilino)phenylacetic acid), and to a lesser extent synthetic estrogens such as 17α-ethinyl-estradiol (EE2, (17α)-19-Norpregna-1(10),2,4-trien-20-in-3,17-diol), widely used in contraceptives, are among the most relevant molecules found in aquatic ecosystems (surface and drinking water), at concentration ranges in the few tens of ng/L [2,16,17,18]. Globally, nearly 25% of the tested drinking water samples contain one to four molecules, and 5% of the treated water samples have a molecule content >100 ng/L [19] (ANSES, Nancy Laboratory of Hydrology 2011, http://www.anses.fr/sites/default/files/documents/LABO-Ra-EtudeMedicamentsEaux.pdf, accessed on 18 March 2011). IBU and DCF have been included in the list of priority hazardous substances by the European Commission (2012). Moreover, 2-hydroxyibuprofen (2hIBU), the main IBU metabolite, has been detected in >5% of treated water samples at levels close to 100 ng/L (ANSES 2011) [19], and may contribute to the environmental risk of these chemicals [20]. The concentrations present in drinking water samples are a thousand to a million times lower than the therapeutic doses. However, this suggests the chronic and uncontrolled exposure of human populations to low doses of multi-component mixtures. Importantly, they could affect the development and function of the reproductive system. Therefore, pregnant women and young children are potentially more sensitive to their effects.

Indeed, epidemiological studies have reported an increase in the occurrence of external sexual system anomalies (hypospadias, cryptorchidism) in newborn boys and of early puberty in girls. These anomalies are the result of the Testicular Dysgenesis Syndrome or Ovarian Dysgenesis Syndrome and may lead to various problems throughout the adult reproductive life [21,22]. The etiology of these syndromes is not completely known but the steady increase in reproductive disorders suggests environmental rather than genetic causes. Particularly, in utero exposure to endocrine disruptor chemicals (EDCs) interferes with the functioning of the male and female endocrine glands [21,23,24,25]. These EDCs affect the development of germ cells, the gamete precursors [26], and therefore can have deleterious inter- and trans-generational effects leading to a reduction in gamete quality and fertility. The critical window of exposure is the onset of fetal gonadal sex determination that coincides with the period of epigenetic reprogramming in developing germ cells [27,28]. Many epidemiological studies on in utero exposure to therapeutic doses of aspirin, IBU, or acetaminophen during the first months of gestation [3,29], and ex vivo/in vivo studies using rodent models [5], identified these molecules as EDCs. In adult men, IBU alters the hypothalamic-pituitary-gonadal axis, leading to a decrease in testosterone production. Moreover, ex vivo studies using adult testis explants confirmed the IBU antiandrogenic effect [30]. EE2 and other estrogenic derivatives have also been identified as EDCs that may affect early development and fertility in exposed rodents and their offspring [31,32] and also human reproductive health [33].

The toxicological risk assessment of environmental doses of IBU, DCF, and EE2 has highlighted their effects on the physiology and reproduction of aquatic species, and dose limits have been defined for these species [34,35]. However, unlike for other molecules present in the environment, quality standards and maximum authorized concentration limits concerning therapeutic drug residues in drinking water have not been established yet (French National Plan for Drug Residues in Waters, 2011), and the health risk for humans is not well known. Therefore, it is urgent to determine the effects of these molecules, present in multi-component mixtures in drinking water, on the reproductive health of model mammalian species to identify putative biomarkers of exposure.

On the basis of the published findings on NSAID and EE2 exposure, we hypothesized that long-term exposure to the IBU, DCF, EE2, and 2h-IBU mixture (at two environmentally relevant doses) could affect the reproductive organ development of in utero exposed F1 mice, leading to an impaired puberty onset. Using cellular and transcriptomic approaches, we studied gonad development and gene expression in the exposed F1 generation and also in the F2 generation (not directly exposed) to evaluate the potential intergenerational effects of these drugs. We identified the Adverse Outcome Pathway (AOP) networks for NSAIDs and EE2 at environmentally relevant doses that may provide new insights for reproductive toxicity studies of other endocrine disruptors based on biomarker transcriptomic analysis.

## 2. Results

### 2.1. Littermates, Body Weight of F1 and F2 Newborns and Postnatal Animals

Two mixtures containing environmentally relevant doses of IBU/2hIBU + DCF + EE2 (dose 1: D1 and dose 2: D2) were added or not (control; C) to the drinking water of pregnant mice and their progeny from 8.5 days post-coitum (dpc) until animal sacrifice (30 days post-partum, dpp, for females and 35 dpp for males) (Appendix A). This exposure period includes early germ cell development, the key periods of somatic and germ cell sex determination, and puberty (7–30 dpp/35 dpp). The number of F1 and F2 pups per litter and their mean weight at 7 dpp were not significantly different between the experimental and control groups and no sign of in utero or neonatal death was observed (*n* = 7 litters for F1–C and F1–D2 and *n* = 6 litters for F1–D1; *n* = 5 litters for all F2 groups except *n* = 4 litters for the CxD1 group) (Appendix A).

At 21 dpp at the time of weaning, the weight of exposed F1 animals (D1 and D2 males and D1 females) was significantly higher than in control animals (Table 1) but not at 35 dpp (males) and 30 dpp (females) (end of puberty) (Table 1). Similarly, in F2 animals, the weight of 21 dpp D1xC males (Table 2) and CxD2, D1xC, and D2xC females (Table 3) was significantly lower compared with control animals, whereas the weight of 35 dpp males and 30 dpp females from other mating combinations was comparable to that of the control animals (Table 2 and Table 3).

### 2.2. Puberty Onset in F1 and F2 Males

The anogenital distance (AGD) is an indicator of the exposure level of androgen during the male programming window (13.5–15.5 dpc) [28,36]. AGD is measured and reported relative to the body weight (BW). The AGD/BW ratio was significantly reduced in 21 dpp F1–D1 and F1–D2 animals compared with controls (Table 1); however, it is recovered in 35 dpp F1 males and was normal in 21 dpp and 35 dpp F2 males (Table 2). These results suggest that androgen production was reduced in F1 male embryos during the male programming window, potentially affecting puberty and postnatal testis maturation.

### 2.3. Puberty Onset in F1 and F2 Females

In females, the AGD/BW ratio was significantly shorter than in males, as expected, and was similar to controls in 21 dpp F1–D1, F1–D2 (Table 1), and F2 animals (Table 3). Vaginal opening, which indicates puberty onset [37], was visually monitored from 20 dpp. In F1 females, vaginal opening started at day 28 dpp in the control and D1 animals, (*n* = 33 Ctrl, *n* = 21 D1) and at day 25–26 dpp in D2 animals (*n* = 28). Vaginal opening was complete (90%) at 39 dpp, 37 dpp, and 36 dpp in the control, D1, and D2 females, respectively. This indicates earlier puberty in F1–D1 and F1–D2 females (2 and 3 days, respectively, compared with controls) (Table 4) and a dose–response effect of the drug cocktail on puberty timing. In F2 females, puberty onset occurred at 28 dpp (CxC, *n* = 11; CxD1, *n* = 15; CxD2, *n* = 35; and D1xC, *n* = 26), 27 dpp (D2xC, *n* = 29), and 26 dpp (D2xD2, *n* = 22) (Table 4), and was complete (90%) at 37 dpp and 34 dpp in D2xC and D2xD2 females, respectively (i.e., 1 and 4 days earlier compared with the other groups) (Table 4). These data indicate that the effect on puberty onset in F1 females is transmitted to the unexposed F2 generation when a F1–D2 male is mated with a control or a D2 female.

### 2.4. Histology (HE) and Maturation of 35 dpp F1 and F2 Testes

Histological analysis (Hematoxylin-Eosin staining) of 35 dpp F1 and F2 testes showed that the seminiferous tubules are formed normally, like in controls, and contained germ cells engaged in the process of spermatogenesis (Figure 1A). However, the tubule diameter was significantly decreased in F1–D2 testes (*p* = 0.0002) compared with controls (Figure 1B); this decrease was also observed in testes from F2 animals obtained from mating F1–D1 males with control females and from mating F1 control males with F1–D2 females (Figure 1B) (D1xC and CxD2 groups, *p* < 0.0001). Moreover, the abnormal presence of undifferentiated germ cells in the lumen of seminiferous tubules (green arrows in Figure 1A) was observed in F1–D2 testes (*p* = 0.029) and in F2 CxD1 (*p* = 0.018), D1xC (*p* = 0.008), D2xC (*p* = 0.0055), and D2xD2 (*p* < 0.0001) testes compared with F1–C and F2 CxC testes, respectively (Figure 1C). Additionally, abnormal tubular features such as disorganized epithelium, vacuolization of the basal compartment (purple arrows), and detachment of germ cells from the Sertoli epithelium (blue arrows), were observed in F1 and F2 testes (Figure 1A), suggesting abnormalities in the seminiferous epithelium and in the blood–testicular barrier (BTB) formation and/or integrity. However, cell proliferation and cell apoptosis, evaluated via PCNA and TUNEL staining, respectively, did not show any significant difference between F1 and F2 testes and their respective controls (Appendix A).

Then, quantification of undifferentiated spermatogonia (SpgA), the precursors of future gametes that express the Lin-28a marker [38], via immunofluorescence (IF) analysis with antibodies against Lin-28a and SOX9 [39], a Sertoli cell marker [40] (Figure 2A), showed that the ratio between the number of SpgA and Sertoli cells which reflects the capacity of Sertoli cells to support spermatogenesis, was significantly decreased in F1–D1 (*p* = 0.0497) and F1–D2 (*p* = 0.0006) testes compared with controls (Figure 2A). This ratio tended to decrease also in F2 testes (Figure 2B). Moreover, IF with an antibody against γH2aX, a marker of DNA double-strand breaks at the zygotene stage of the prophase I of meiosis [41], showed that meiosis progressed normally in F1–D1 and F1–D2 testes and in F2 testes compared with their respective controls (Figure 2C). In F2 D1xC, D2xC, and D2xD2 testes, some γH2aX-positive cells were identified in the center of some tubules, suggesting that DNA damage occurred in these cells (arrows in Figure 2C). The density of round spermatids produced in F1–D1 and F1–D2 testes was significantly reduced compared with controls (*p* = 0.0068 and *p* = 0.0003, respectively) and also in F2 testes from CxD2 (*p* = 0.0065), D1xC (*p* = 0.0077), D2xC (*p* = <0.0001), and D2xD2 (*p* < 0.0001) animals (Figure 2D).

Sertoli cells are somatic cells involved in supporting SpgA differentiation during spermatogenesis [40]. Analysis of the expression of anti-Müllerian hormone (AMH), a member of the TGFb superfamily [42] and a marker of immature Sertoli cells [43], showed that it was still expressed in 35 dpp F1–D1 and F1–D2 testes but not in control testes, suggesting a delay in Sertoli cell maturation after drug exposure, and was also expressed in F2 CxD2, D2xC, and D2xD2 testes (Figure 3A). IF analysis of the nuclear receptor chicken ovalbumin upstream promoter-transcription factor II (COUPTFII, also known as Nr2f2), which is involved in Leydig progenitor formation and differentiation and Leydig cell maturation [44], indicated that the number of Leydig cells was increased in the hyperplasic interstitial space of F2 testes compared with the controls (CxC group) but not in F1 testes (Figure 3B).

The expression of *Amh* and *Nr2f2* (*CouptfII*) genes tended to increase, and *Lin28a* and *Sox9* expression tented to decrease in F1–D1 and F1–D2 testes compared with the controls (Appendix A); however, gene expression levels were not significantly different compared with the controls.

As Sertoli cell function and interactions with the germ cells depend on the integrity of the blood–testicular barrier (BTB) [45], a biotin assay was used to investigate whether exposure to these molecules affected the BTB structure. In F1 control and D1 testes, a biotin signal was observed in the basal membrane of the tubules but not in the seminiferous tubules (Figure 4A). The biotin signal slightly penetrated into the seminiferous tubules in D2 testes (Figure 4A), suggesting that BTB integrity was impaired. Connexin 43 (Cx43), a constitutive gap junction protein that interacts with tight and adherens junction proteins, is crucial for spermatogenesis [46]. Staining with an anti-Cx43 antibody shows in D2 testis an irregular organization of Cx43 expression with several spots and dotted lines within the tubules, compared with the basal expression in the region of the BTB in control or D1 testes (Figure 4B), suggesting that the BTB might be destabilized. The Cx43 expression pattern in D2 testis is also shown in F2 C–D2, D1–C, D2–C, and D2–D2 testis (Figure 4B). Next, as the cytoskeleton is essential for cell junction maintenance in the BTB [46], a-tubulin organization was analyzed. In stages IV–VII tubules of the control testes, tubulin staining was located in the cytoplasm of Sertoli cells and in the apical zones that surround developing spermatids. Its localization was disturbed in F1–D1 and F1–D2 and also in F2 testes, as indicated by the presence of tubulin around spermatocytes and round spermatids (Appendix A). This suggested that exposure to these drug cocktails might impair BTB integrity and the cytoskeleton organization in an intergenerational manner. These alterations might explain the abnormal localization of some germ cells in the F1 and F2 seminiferous tubules (Figure 1A–C).

To evaluate the spermiogenesis onset [47], chromomycin A3 (CMA3) that binds to protamine-free chromatin [48] was used to evaluate the level of protamine deposition on elongated spermatids (eSpd) (stage 16) (Figure 5A). The mean number of stages VII–VIII tubules that contained CMA3-positive eSpd was significantly decreased in F1-D1 (*p* = 0.0041) and -D2 (*p* = 0.0093) testes compared with controls (Figure 5B), and also tended to decrease in F2 CxD2, D1xC, D2xC and D2xD2 testes (not significant). This finding suggests that spermiogenesis onset may be advanced by exposure to this drug cocktail (both doses).

### 2.5. Histology and Folliculogenesis Onset in 30 dpp F1 and F2 Ovaries

Histological analysis of 30 dpp F1 and F2 ovaries showed a similar follicular organization in exposed and control ovaries. They contained typical populations of primordial, primary, secondary, and antral follicles (ovulation does not occur at this stage) (Figure 6). However, follicular cell density within growing follicles seemed to be lower in F1 and F2 ovaries compared with controls (Figure 6, enlarged panels). IF analysis with antibodies against FOXL2, a granulosa cell marker [49,50], and alpha-smooth muscle actin (a-SMA), showed that follicles contained well-differentiated granulosa cells (FOXL2^+^) and that follicles were well-delineated via a-SMA staining (Appendix A). Moreover, the density of granulosa cells (FOXL2^+^), evaluated by counting their number in regions of interest (ROI) (*n* = 4–8 ROIs, *n* = 3–5 animals per group) was decreased in F1–D1 (*p* < 0.0001) and F1–D2 (*p* < 0.0001) and in F2 D1xC, D2Xc, and D2xD2 (*p* < 0.0001) ovaries compared with their respective controls (Figure 7A,B). TUNEL staining showed that apoptosis was similar in F1 and F2 ovaries and controls (Appendix A). Granulosa cell proliferation, measured by the S-phase marker phospho-histone H3 (PH3), was reduced in F1–D1 (*p* = 0.0184) and F1–D2 (*p* = 0.0364) ovaries, but was not affected in F2 ovaries (*n* = 3–5 animals per group) (Appendix A, Figure 7C).

In mice, the primordial follicle (PF) pool is established during the neonatal period and represents the oocyte resource and follicle reserve during reproductive life [51,52]. The ratio between the number of primordial follicles and the total number of follicles in F1–D1 and F1–D2 ovaries tended to decrease, but not significantly. However, this ratio was significantly reduced (*p* =0.0246) in F2 D2xD2 ovaries (*n* = 4) (Figure 6), suggesting a partial depletion of the primordial follicle pool. AMH is expressed in the granulosa cells of growing follicles from puberty to the adult stage [53] and inhibits primordial follicle activation and preantral follicle-stimulating hormone (FSH)-stimulated follicle growth [54]. The number of AMH^+^-growing (primary/secondary) follicles was significantly decreased only in F2 D2xD2 ovaries (*p* = 0.0436). Moreover, the percentage of antral follicles still expressing AMH was significantly increased in F1–D1 (*p* = 0.0234) and F1–D2 (*p* = 0.0568) and also F2 D2xD2 (*p* = 0.0436) ovaries (*n* = 3–4 animals per group; Appendix A).

COUPTFII is expressed in theca cells that surround growing follicles where it activates the expression of steroidogenic genes [55,56]. Overall, COUPTFII expression was similar in F1 and F2 and the control ovaries (Appendix A). In control F1 (Ctrl) and F2 (CxC) ovaries, COUPTFII staining was found in two to three cell layers surrounding the granulosa cells of primary and secondary follicles. Staining was stronger in four to five layers of elongated cells surrounding antral follicles. Weaker staining was also detected in ovarian stromal cells (Appendix A). However, in F1–D1, F1–D2 and in F2 CxD2, D1xC, D2Xc, and D2xD2 ovaries, preantral follicles contained disorganized theca cell layers: some cells with round nuclei, losing their elongated shape (Appendix A). Furthermore, in F1–D2 and in F2 D1xC, D2xC, and D2xD2 antral follicles, the number of theca cell layers was decreased, and cells were disorganized (Appendix A).

Altogether, these findings suggested that functional cell interactions between somatic cells and potentially cell communication with oocytes might be affected. Gap junctions that are established in the primordial follicle and remained throughout the follicular growth are necessary for follicle development and oocyte growth [57]. Cx43, the main gap junction protein that forms channels between the granulosa cells [58], was detected as a strong punctuate staining at the borders of the granulosa cells in preantral and large antral follicles of all F1 and F2 ovaries (Figure 8A). In the control ovaries, weak Cx43 staining was also weakly detected in the zona pellucida around the oocytes in large follicles, where it allows oocyte–somatic cell communication through the trans-zonal projections (TZPs) [59,60]. Its expression was increased in the TZP and zona pellucida between the granulosa cells and oocyte surface in F1–D1 and D2 and F2 ovaries compared with their respective controls (Figure 8A, enlarged panels). Microtubules, the cytoskeleton components, are involved in maintaining cell shape and are organized into bundles running parallel with the long axis of granulosa cells [61]. In control F1 and F2 ovaries, a-tubulin was homogeneously expressed around the granulosa cells of antral follicles (Figure 8B). Its expression became irregular in F1–D1 and F1–D2 and in F2 D1xC, D2Xc, and D2xD2 ovaries, suggesting cytoskeleton disorganization leading to a modification of the granulosa cell shape. Thus, the intercellular communications between granulosa cells and oocytes, and the extensive remodeling that occurs during follicular development, might be disturbed in an intergenerational manner upon exposure to these drug cocktails.

### 2.6. LH, FSH, Testosterone, Estradiol Quantification in Serum

In F1–D1 and F1–D2 males, the mean testosterone concentration in serum tended to be lower than in the controls (2–1.5 ng/mL in D1/D2 and 4 ng/mL in controls). Similarly, the testosterone concentration in serum samples from F2 CxD1 and CxD2 (exposed mother), D1xC (exposed father), and D2xD2 (both parents exposed) males tended to be lower (30–60 pg/mL) than in the controls (Figure 9A). However, these differences were not significant (*p* >0.05). Serum follicle-stimulating hormone (FSH) (Figure 9B) and luteinizing hormone (LH) (Figure 9C) concentrations were not significantly different between the F1 and F2 males and their controls. However, the gonadotropin concentration tended to be lower in F1–D1 (mean: 0.15 ng/mL of LH and 0.5 ng/mL of FSH) and higher in F1–D2 (0.35 ng/mL of LH and 3.2 ng/mL of FSH) males compared with the controls (0.90 ng/mL of LH and 1.8 ng/mL of FSH). In F2 CxD1 and CxD2, D2xC, and D2xD2 males, LH (3.2–5 ng/mL) and FSH (1.9–3.2 ng/mL) concentrations tended to be higher than in controls (2.5 ng/mL of LH and 1.2 ng/mL of FSH). These results suggest that the hypothalamic–pituitary–gonadal axis is not significantly disturbed by exposure to these drug cocktails.

In females, the E2 concentration was not significantly different between F1–D1 and F1–D2 and their controls (7–10 pg/mL). In F2 CxD1, CxD2, D1Xc, and D2xD2 females, the E2 concentration tended to be lower (10 pg/mL) than in the controls (20 ng/mL) (Figure 9D).

### 2.7. Differential Gene Expression in Pubertal F1 Exposed Testis and Ovary

RNA-seq analysis of biological triplicates with a depth of 45 to 163 million reads per library using an Illumina sequencer (flow cell S4, 100 bp, paired-end reads) was used to determine the gene expression in F1 testes and ovaries (Appendix A). The numbers of differentially expressed genes (DEGs) ranged from 3 to 167 in F1 males, 56 to 535 in F2 males, 16 to 39 in F1 females, and 6 to 209 in F2 females, and included upregulated (from 33.3% to 69.7%) and downregulated (from 30.3% to 57.9%) genes (Appendix A). In samples from F1 D1 + D2 males, 36 DEGs (21.1%) were RIKEN cDNA sequence and 33 DEGs (19.4%) were pseudogenes and non-coding RNAs. Conversely, in samples from F1 D1 + D2 females, DEGs contained only one RIKEN cDNA sequence (Appendix A). In F1–D1 males, 7/30 downregulated genes were downregulated by more than 10-fold and 6/69 upregulated genes were upregulated more than 10-fold (particularly *Pga5* that encodes an aspartic protease and was upregulated by 287-fold). In F1–D2 males, two major DEGs were downregulated by 6.6-fold (*Tap1*) and upregulated by 12.2-fold (*P2rx5*), respectively. In females, the mean expression change was lower than in males, from 0.44- to 1.63-fold in F1–D1 and from 0.61- to 2.97-fold in F1–D2 samples (Appendix A).

Gene Ontology (MGI GO tool) showed that 34.6, 49.5, 42.5, and 29.7% of the F1 D1 + D2 male DEGs were related to the immune system/response to stress (*p* < 10^−7^), cytoskeleton/acting binding (*p <* 10^−4^), metabolic process (*p <* 10^−2^), and development/morphogenesis (*p <* 10^−2^) GO terms, respectively (Appendix A). In females, 55.5, 38.9, 20.3, 35.1, and 11.1% of F1 D1 + D2 DEGs were related to the metabolic process (*p <* 10^−6^), development/morphogenesis (*p <* 10^−5^), reproductive process/sex differentiation (*p <* 10^−6^), cell growth/apoptosis (*p <* 10^−6^/10^−5^), and immune system/response to stress (*p <* 10^−3^) GO terms, respectively (Appendix A).

In males, exposure to this drug cocktail led to the activation of some inflammation-promoting genes and to the downregulation of some inflammation suppressors (Figure 10A, Appendix A), suggesting that exposure may induce an immune response within the testis. Genes encoding the actin-binding cytoskeleton (such as *Actn3*), microtubule-binding, cell adhesion (such as *Emilin2*, *Cntnap5b, Pdpn*, *Dpep1)*, tight junction, and extracellular matrix (ECM) (such as *Col5a1, Impg2*, *Itga2)* proteins also were affected, confirming that cytoskeleton remodeling/actin binding were modified in F1 testes (Figure 10B, Appendix A). Moreover, *Pga5* was strongly overexpressed in F1–D1 testes (x287) but not in F1–D2 testes (Figure 10C, Appendix A). In the ovary, *Pga5* encodes an aspartic acid protease that is expressed in the granulosa cells of the preovulatory mouse follicle [62] and before the follicle rupture [63]; it is also involved in the degradation of maternal placental proteins [64]. Similarly, other serine proteases, *Klk1*, *Klk1b3*, and *Prss56*, were significantly upregulated only in F1–D1 testes [65] (Figure 10C, Appendix A). Thus, these proteases with other peptidases (Pga5, Dpep1), and matrix metalloproteinase (MMP) family members [66], may increase ECM remodeling during testis maturation.

The increased inflammation and the cytoskeleton and ECM alterations in exposed testes may explain the observed phenotypes and may alter testis physiology. Indeed, the expression of many genes involved in regulating metabolic and secretion processes was also affected (Appendix A). This group included genes encoding proteins involved in lipid homeostasis, fatty acid metabolism and transport, and steroidogenesis, such as fatty acid-binding protein, *Fabp3* and *Fabp4*, *Lpl*, *Hsd3b6* (a major gene implicated in steroid biosynthesis), *Edh3* (participating in testosterone biosynthesis) [67], and *Dhrs4* (involved in retinal metabolic process) [68]. *Amd2* (encoding S-adenosylmethionine decarboxylase 2, a limiting enzyme in the biosynthesis of polyamine/spermine/spermidine [69] that are involved in many biological processes, such as cell proliferation and apoptosis, aging and reproduction [70]) was also downregulated in F1–D1 testes. These molecules bind to various cation channels. Some genes encoding channel proteins, particularly *Kcnk2* (potassium channel; expressed in bovine sperm) [71] and *Trpv1* (transient receptor potential cation channel) [72] were also downregulated in F1–D1 testes. Moreover, several genes involved in developmental processes are differentially expressed, particularly *Foxa3* (transcription factor that is a negative regulator of steroidogenesis in Leydig cells [73] and of the serine proteases kallikreins [74]), and *Sall1* (spermatogonial cell marker involved in spermatogonia stem cell (SSC) self-renewal and differentiation [75]). Sall1 downregulation in exposed testes might affect spermatogonia stem cell differentiation into spermatogonia, as observed in F1–D1 and F1–D2 testes.

In females, more than 50% of DEGs in F1–D1 and F1–D2 ovaries were related to metabolic processes, retinoic acid, fatty acids, phospholipids, and cholesterol binding/metabolism/biosynthesis (*p* < 10^−6,^
Appendix A). Additionally, many DEGs were involved in ECM, cytoskeleton organization, actin polymerization/binding, and tight junctions (Figure 10D, Appendix A). Furthermore, genes involved in promoting immune responses were downregulated, and genes involved in inhibiting immune responses were upregulated (Appendix A), suggesting that exposure to these drugs tends to lower the inflammatory response in ovaries, unlike in testes.

In exposed females, DEGs also belonged to development and morphogenesis and sex differentiation and reproductive processes (*p* < 10^−6^/10^−5^, Appendix A, Figure 10E). The interaction between ovarian cells may be impaired because expression of TGF-b-related factors (*Bmpr2*, *Fst*, *Gdf10* (*Bmp3*)) and of Wnt signaling components (*Sfrp1* and *Bcl6b*) [76] was affected in F1–D2 ovaries. The Wilms’s tumor gene *Wt1,* which activates *Fst* (follistatin) during follicle development, was also downregulated in F1–D1 ovaries. Moreover, genes implicated in germ cells were downregulated in F1–D1 ovaries: *Zp1* and *Zp2*, which are the main components of the zona pellucida around oocytes and are involved in folliculogenesis [77] and in oocyte growth and fertilization [60], *Padi6,* which regulates microtubule-mediated organelle positioning in oocytes [78], and is required for female fertility [79] and *Tmem184a* (*Sdmg1*), are significantly down regulated in F1–D1 ovaries. Analysis of *PadI6*, *Zp1*, *Zp2* and *Nlrp14* expression via a RT-qPCR in F1–D1 and F1–D2 oocytes confirmed these results (Figure 10F). The expression of several genes involved in primordial follicle activation [80] was modified in F1–D1 ovaries, such as *Nlrp4f* and *Nlrp14*, that are downstream of *Figla* and *Sohlh2*, respectively, in the promotion of secondary follicle oocyte growth [81,82]. *Tcf21* (*Pod1*) was upregulated in D2 ovaries. This factor may negatively regulate Leydig cell differentiation and steroidogenesis through repression of the nuclear receptor NR5A1 [83] and may inhibit the expression of some MMPs for the regulation of cell–matrix compositions [84].

Altogether, these data suggest that the altered expression of genes implicated in metabolism, development and morphogenesis, extracellular matrix/cytoskeleton organization, inflammation, and folliculogenesis regulation in ovarian cells may explain the modified ovarian physiology observed in exposed ovaries and the early puberty of exposed females.

### 2.8. Biomarkers of Exposure: Genes That Are Differential Expressed Also in the F2 Generation

For F2 samples, the depth of sequencing ranged between 43 and 113 million reads per library (Appendix A). The numbers of identified DEGs ranged from 56 to 535 in F2 testes and from 6 to 209 in F2 ovaries, with similar numbers of upregulated and downregulated genes in each sex (Appendix A). The mean gene expression change ranged from 588-fold downregulated to 526-fold upregulated (Appendix A) in males, and from 66.6-fold downregulated to 96-fold upregulated in females (Appendix A). Gene Ontology (MGI GO tool) identified similar enrichment pathways (metabolic process, development/morphogenesis, cytoskeleton/acting binding/cell adhesion, immune system/response to stress) as described for the F1 samples (Appendix A).

Comparison of F2 DEGs and F1 DEGs to identify genes that remained differentially expressed also in the F2 generation showed that 13 (testes) and 11 (ovaries) genes were differentially expressed in both generations (Appendix A). Among these conserved DEGs, the expression level of *Amd2* (spermine/spermidine biosynthesis [69] was similarly modified (~two-fold decrease) in F1 and in F2 CxD1 and D2xC testes. It was also downregulated in F2 CxD2, D1xC, and D2xD2 ovaries (Appendix A). *Pga5* was similarly modified in F2 testes (dd7368 in CxD2 and ×526 in D1xC samples) and in F1–D1 testes (×287). *Pga5* encodes an aspartic acid protease that might be involved in ECM remodeling with the serine peptidase *Prss56* that was upregulated in F1 and also in F2 CxD2 testes. Other genes involved in ECM/cytoskeleton organization/microtubule binding, such as *Actn3* (alpha-actinin-3)*, Emilin2* (elastin microfibril interface-located protein 2)*, Cntnap5b* (contactin-associated protein like 5-2), and *Tcp10c* (T-complex protein 10c)*,* were deregulated in both F1 and F2 testes and ovaries. Particularly, *Emilin2* was downregulated in F1–D1 and in F2 D1xC testes, whereas it was upregulated in F2 ovaries. Conversely, the inflammation suppressor *Cfd* was downregulated in F1–D1 testes, and was strongly upregulated in F2 CxD2, D1xC, and D2xC ovaries. Similarly, *Fabp4* and *Fabp3* were downregulated in F1–D1 testes and upregulated in F2 D1xC (*Fabp4* ×3.54, *Fabp3* ×0.69) and D2xC (*Fabp4* ×2.86) ovaries. Fabp3 mediates lipid transport and accumulation in maturing oocytes, and colocalizes with TZPs around the oocyte [85]. Conversely, DEGs identified in F1 ovaries were detected only in F2 ovaries: *Rnf19b* (in F2 CxD2) and *Crapb2* (in F2 D1xC).

These data suggest that exposure to NSAID–EE2 cocktails affects the expression of key genes involved in ECM organization and remodeling (*Amd2*, *Pga5*, *Actn3*, *Emilin2, Cntnap5b*), in inflammation (*Cfd*, *H2-T10*, *H2-T23*), and in metabolism (*Fabp3*, *Fabp4*) in an intergenerational manner. These genes, particularly *Amd2*, *Emilin2*, *Pga5*, and *Fabp4* that were altered in F2 animals with an exposed father, might be considered as biomarkers of exposure.

### 2.9. Putative Markers of NSAIDs and EE2 Exposure That Are Also Markers of Exposure to Other EDC

Comparison of these DEGs with those identified after exposure to other EDCs indicated that 1 to 17 DEGs detected in F1 testes were also deregulated after exposure to the xenoestrogens bisphenol A (BPA) [86] and EE2 [87] and to the anti-androgenic EDCs di-2-ethylhexyl phthalate (DEHP) [88], DDT, and vinclozolin [89] (Appendix A). Some genes that were also differentially expressed in F2 testes might be biomarkers of exposure to estrogenic (*Actn3*, *Cntnap5b, Kcnk2*, *Sall1*) or anti-androgenic (*Dhrs4*, *Fabp3*) compounds.

Comparison of the 55 DEGs identified in F1–D1 and F1–D2 ovaries with those identified after exposure to other EDCs showed that 32 (61%) were shared (Appendix A). Specifically, *Arhgap29* and *Cyfip2* (genes involved in regulating ECM and actin filament organization) were deregulated also after exposure to bisphenol S (BPS) [90], *Bicd1* (microtubule cytoskeletal organization), and *Cyfip2*, *Nlrp4f*, *Crabp2*, and *Rnf19b* (actin filament organization) were deregulated also after exposure to pentachlorobiphenyl (PCB) [91]. *Nlrp4f*, *PadI6*, *Tcf21* (ECM/actin/microtubule organization), *Zp2* (the transcriptional repressor in the circadian system that targets *Cyp19a1* expression and inhibits estrogen synthesis (*Nr1d1*)) [92], and *Nlrp14* (oocyte and spermatogenic marker) [93] were upregulated also after exposure to deoxynivalenol (DON). DON is an environmental toxin that frequently contaminates cereal crops and food and disrupts estrogen and progesterone secretion in murine granulosa cells [94]. *Crabp2*, *Crip1*, *Dhcr4*, *Fst,* and *Sfrp1* were also downregulated after exposure to DON. Some ovarian DEGs (*Zp1*, *Zp2*, *PadI6*, *Nlrp14*, *DNAja1*) were also differentially expressed after exposure to DEHP [95] and pesticides (Vincozolin, DDT) [96]. *Dnaja1*, *Gstm2*, *Nlrp14*, *Nlrp4f*, *PadI6*, *Zp1*, and *Zp2* are particularly sensitive to EDC exposure, suggesting that these genes might be biomarkers of ovary exposure.

### 2.10. F1 DEGs Related to Human Diseases

Among the DEGs identified in F1 testes, *Foxa3* and *RNAse L* are implicated in reproductive diseases (Appendix A). The phenotype of *Foxa3*^−/−^ mice is similar to the human spermatogenic atrophy, which is characterized by a variable degree of seminiferous tubule degeneration [74]. In humans, FOXA3 is expressed in Leydig cells and is induced by LH/cAMP signaling. Its overexpression suppresses the cAMP-induced expression of NUR77 and its target steroidogenic genes, resulting in decreased production of testosterone [73] and fertility defects [97]. Expression of *Foxa3* in the testis is also modified upon exposure to DEHP [98]. The endoribonuclease RNAseL (male F1 DEG ×0.43) plays a role as a tumor suppressor, and polymorphisms in this gene are involved in prostate cancer [99]. Moreover, RNAseL functions within a network of innate immune proteins that are associated with cytoskeleton components to serve as a barrier to pathogen infection [100]. Several male DEGs have been implicated in immune diseases, particularly genes of the MHC class I and II regulation (*Tap1*, *Ciita*, *H2-Q4*, *H2-T-ps*, *H2-T10* and *H2-T23*). *DNAJC19* is a major component of the translocation machinery of mitochondrial membranes that is involved in cell growth and metastasis [101]. *HSD3B6* is specifically expressed in adult Leydig cells to regulate steroidogenesis [102]. *LPL* is involved in fatty acid synthesis, is downregulated in Sertoli cell-only and round spermatid maturation arrest syndromes [103], and is expressed in germinal cell tumors with embryonic features [104].

Several DEGs identified in F1 ovaries are related to human reproductive diseases, particularly *DHCR24* (prepubertal growth retardation and adult infertility), *DNAJA1* (defective androgen regulation in Sertoli cells leading to spermatid defects), *FST* (presence of male-specific coelomic vessels in ovary and loss of oocytes), *TCF21* (infantile uterus, male infertile), *Wt1* (somatic and germ cell lineages), *ZP1* (defects in fertilization), and *ZP2* (defects in oocytes in developmental competence) [97]. In addition, the deregulation of *GSTM2* (DNA damage), *SFRP1*, and *BMPR2* (involved in folliculogenesis) has been described in ovarian diseases [105] (Appendix A). Among these genes, *ZP1*, *WT1*, and *BMPR2* are premature ovarian insufficiency (POI) genes in humans [106]. Moreover, besides *Bmpr2*, *Fst*, and *Tcf21* are deregulated in an AMH-induced mouse model of polycystic ovarian syndrome (PCOS) [107] (Appendix A).

## 3. Discussion

Drug use has largely increased in the past decades due to their beneficial effects and population growth. Consequently, they are now major environmental contaminants that represent a source of chronic exposure for wildlife (effluent discharge sites) and for human populations (drinking water) [2]. Pregnant women are widely exposed to a multitude of environmental chemicals that may affect reproductive organ development and function in adult life [21,23,24,25]. Indeed, the hypothesis of a “fetal origin of adult disease” [108] suggests that besides genetics causes, environmental factors that affect early developmental stages increase the risk of developing some diseases later during adult life.

This study evaluated for the first time, at the cellular and transcriptomic levels, the effects of chronic exposure to a mixture of IBU, 2hIBU, DCF, and EE2, at two environmentally relevant doses, on postnatal testes and ovaries in mice, and explored their potential intergenerational impacts. The chosen drugs are among the most relevant molecules found in drinking water and were used at doses that are relevant to everyday human exposure. Critical periods of early gonad development (i.e., proliferation and differentiation of supporting cells and germ cells, the precursors of gametes in adult life [109]) are tightly regulated and are very sensitive to any perturbation induced by environmental chemical exposure [21].

We found that, at the end of puberty and before sexual maturity, testes and ovaries from exposed F1 mice were overall morphologically normal. Seminiferous tubules were well-organized and contained germ cells engaged in spermatogenesis. Ovaries contained well-defined follicles, but not polyovular follicles that are classically observed after exposure to higher doses of EE2, DES, genistein, or other estrogenic-like compounds [110,111]. A more thorough histological analysis highlighted the presence of alterations (summary in Figure 11) in F1 testes and ovaries and also in the F2 progeny when the F1 father was exposed. These phenotypes might be the consequences of the ECM (i.e., connexin 43 and a-tubulin) disorganization detected via immunostaining in supporting cells and germ cells in F1 exposed testes and ovaries. In addition, exposure to the drug cocktail significantly altered the expression of genes implicated in cytoskeleton organization, actin polymerization, microtubule-binding, and ECM remodeling (biomarkers in Figure 11). These modifications may affect cell shape maintenance and might impair interactions between somatic cells and germ cells (spermatocytes/spermatids and oocytes) [45,59,112,113]. They may also partly explain the observed phenotypes in testes (abnormal localization of germ cells in tubules, impaired BTB, delayed Sertoli cell maturation, decreased SpgA/Sertoli cell ratio, and round spermatid pool) and ovaries (decreased granulosa cell proliferation, disorganized theca cell layers) (Figure 11).

Other EDCs such as phthalates, bisphenols, perfluorooctanesulfonic acid (PFOS), and also heavy metals (chrome or cadmium) affect the cytoskeleton and gap junction organization, leading to BTB destabilization in males [114] and altered ECM remodeling in females [115]. Moreover, exposure to NSAIDs and EE2 affected the expression of pro-inflammatory and inflammatory suppressors genes, modifying immune responses in the testis and the ovary. Besides the BTB, Sertoli cells also display essential immunoprotective functions that contribute to protect germ cells. These signaling molecules play crucial roles in normal testis development and function, but their upregulations can affect Leydig cell steroidogenesis and spermatogenesis [116], and increased inflammation in the testis has been associated with infertility in humans [117]. Our data suggest that the impaired BTB and cytoskeleton organization in exposed testes and ovaries might be directly related to this modified inflammatory environment.

Moreover, the AGD, an indicator of in utero exposure to testosterone during the male programming window [28,36], was decreased in 21 dpp F1 males but was normal in 35 dpp animals. This suggests a decreased production of testosterone during the embryonic stages that may delay the puberty onset. This significant but mild effect may be due to the low environmental concentrations of the mixture. Indeed, exposure to therapeutic doses of NSAIDs or acetaminophen leads to a decreased AGD in rodent models [3,118,119]. Conversely, in females, puberty occurred earlier in F1 and also in F2 D2xD2 females. This is consistent with data on exposure to estrogenic-like compounds such as bisphenols, BPA [120,121], the BPA analogs BPS and BPE [122], and EE2 [123,124]. Major genes implicated in development and morphogenesis processes, such as the *Wt1* (follicle development), *Tcf21* (negative regulator of steroidogenesis) [83], components of the Tgfb (*Bmpr2*, *Fst*, *Gdf10*) and Wnt (*Sfrp1*, *Bcl6b*) signaling pathways, and genes involved in granulosa–oocytes cell interaction (*Zp1*, *Zp2*, *Padi6*, *Nlrp14*) were affected by exposure to the drug cocktail and may explain the female phenotype. EE2 exposure can alter gametogenesis and epigenetic regulations during early development [31,32], and this might explain the effect of exposure to low concentrations of EE2 on the puberty onset of F1 females and F2 D2xD2 (both parents exposed to the D2 dose). In humans, epidemiological studies suggest a link between alterations in puberty timing and a wide range of adverse health outcomes in adult life, particularly exposure to EDCs [125].

On the other hand, spermatogenesis in males and folliculogenesis in females progressed normally. This may be due to the low concentrations of chemicals used in this study and to variability in the responses to chemicals among mice. However, hyperplasia of the interstitial steroidogenic compartment was observed in F2 but not in F1 testes, and theca cell disorganization in F1 and F2 ovaries. This suggests that the number of steroidogenic cells and steroidogenesis may be affected by other factors than those linked to the hypothalamic–pituitary–gonadal axis, or that gene expression deregulation might have been compensated. In males, concomitant upregulation of *Hsd3b6*, a major gene of steroid synthesis, and of *Foxa3*, a negative regulator of Nur77, a key steroidogenic regulator [73] and of the ESR1 pathway activation [126], and downregulation of *Edh3* (NR5a1-activated gene involved in testosterone synthesis) [67], may affect Leydig cell function. Leydig cell hyperplasia has been classically observed after exposure to EDCs, such as acetaminophen and IBU [127], phthalates [128], and nitrophenol [129].

Both NSAIDs and EE2 may have contributed to the changes observed in F1 testes and ovaries. Estrogens exert relevant roles in gonadotropin and testis physiology in mice and humans [130], and in utero exposure to BPA or other estrogen-like chemicals affects male reproductive developmental outcomes [131,132]. Moreover, the effects of NSAID exposure on female reproduction are well documented [5,133]. Altogether, these data suggest that this disturbed testicular and ovarian cell microenvironment may explain their altered maturation and physiology, and ultimately the early puberty of exposed females. Future experiments will determine whether and how these physiological and molecular modifications affect the reproductive health of adult animals. Particularly, early puberty may be followed by PCOS in adult life, as suggested in humans [134].

Most of the phenotypes we observed in exposed F1 gonads were also identified in F2 gonads, suggesting that like other EDCs, NSAIDs and EE2 have intergenerational effects, as described earlier [31,32,127,133]. Comparison of the F1 and F2 gonadal transcriptomes showed that the expression levels of key genes involved in ECM organization and remodeling (*Amd2*, *Pga5*, *Actn3*, *Emilin2*, *Cntnap5b*), in inflammation (*Cfd*, *H2-T10*, *H2-T23*), and in metabolism (*Fabp3*, *Fabp4*) were modified in an intergenerational manner. These genes, particularly *Amd2*, *Emilin2*, *Pga5*, and *Fabp4* in F2 animals with an exposed father, might be considered as biomarkers of exposure to these mixtures during embryonic and neonatal development (Figure 11).

Furthermore, several DEGs in F1 testes have been previously described as deregulated upon exposure to other EDCs (BPA, phthalates, vinclozolin). Therefore, they might be potential biomarkers of exposure to estrogenic (*Actn3*, *Cntnap5b, Kcnk2*, *Sall1*) or anti-androgenic (*Dhrs4*, *Fabp3*) chemicals in males. Similarly, 61% of DEGs in F1 ovaries have been described also after exposure to other EDCs. The germ cell genes *Dnaja1*, *Gstm2*, *Nlrp14*, *Nlrp4f*, *PadI6*, *Zp1*, and *Zp2* are particularly sensitive to EDC exposure, suggesting that these genes might be biomarkers of ovary exposure to EDCs, including NSAIDs and EE2 (Figure 11).

In addition, some DEGs in F1 testes were related to human diseases, such as reproductive (*Foxa3*, *Dnajc19*, *Lpl*, *Hsd3b6*) or immune (*Tap1*, *Ciita*) disorders. Particularly, FOXA3 plays critical roles in metabolic homeostasis, and is involved in the progression of non-alcoholic fatty acid liver disease [135], and in adipocyte differentiation in obese patients [136]. In mice, *Foxa3* is essential for germ cell maintenance and fertility [74]. Deficiency of HSD3B2, the human homolog of HSD3B6, leads to disorders of sex development [137,138,139]. Together with the hyperplastic phenotype of the F1 and F2 testis interstitial compartment, this suggests that exposure to NSAIDs and EE2 affects Leydig cell physiology. On the other hand, CIITA (the class II major histocompatibility complex transactivator) defects are associated with bare lymphocyte syndrome type II, a severe human immunodeficiency syndrome that could modify the inflammation regulation [140].

The incidence of POI and PCOS has increased in recent decades [25,141]. These disorders affect the fertility and quality of life of young women. Some DEGs in F1 ovaries are related to reproductive/ovarian diseases, particularly POI (as *Wt1*, *Bmpr2*, *Zp1*, *Zp2*) and PCOS (as *Tcf21*, *Fst*, *Bmpr2*) (Figure 11). Deficiency of *WT1*, a sex-determining gene, induces ovarian insufficiency in humans through the regulation of granulosa cell differentiation during follicle development [142,143]. Moreover, BMPR2 and its ligand BMP15 are associated with or are the cause of POI [144]. Similarly, alteration of BMPR2 gene expression can lead to a polycystic ovary in mice without hyperandrogenism [145].

## 4. Materials and Methods

### 4.1. Animal Housing, Experimental Design, and Exposures

For this project, time-mated CD-1 females (*n* = 6) were purchased from Charles River Laboratories (Les Oncins, Saint Germain Nuelles, France) and shipped at gestational day 5. Their pups constituted a cohort of six groups of F0 animals that were weaned and separated at 21 dpp to form six groups of F0 adult animals of each sex. Animals were reared in polyphenylsulfone (PPSU) cages in controlled environmental conditions (light/darkness: 12 h/12 h, 23 °C) (Tecniplast X-Temp, Tecniplast, Décines-Charpieu, France), and drinking water was put in PPSU bottles wrapped in aluminum foil. Animals were fed SAFE D131 (Safe-Diets Company, North Little Rock, AR, USA) that does not contain fish proteins (presence of xenoestrogens, pesticides), soy (rich in phytoestrogens), or alfalfa (presence of isoflavones). Municipal tap water was provided ad libitum. Before starting the exposure protocol, the water quality was tested (CARSO Laboratory, Vénissieux, France) to confirm the absence of some chemical contaminants (pesticides, phthalates, bisphenols, and pharmaceuticals), including IBU, 2h-IBU, DCF, and EE2. All animal experiments were carried out in accordance with procedures approved by the Réseau des Animaleries de Montpellier (RAM) (approval number 34–366 for B.B.-B. animal experimentation) and by the regional ethics committee. Mice were euthanized via cervical dislocation in accordance with the recommendations for animal experiments. The 3R rule was respected throughout this project.

F0 males and females from different litters were mated to minimize inbreeding and mated females were divided into three groups: control (no exposure), IBU/2hIBU + DCF + EE2 dose 1 (D1), and IBU/2hIBU + DCF + EE2 dose 2 (D2) (*n* = 6–7 females per exposure group) (Appendix A). The IBU, 2hIBU, DCF, and EE2 doses were calculated on the basis of their mean concentrations (5, 40, 10, and 1 ng/L, respectively) and maximum concentrations (50–100, 85–100, 50, and 20–50 ng/L, respectively) found in environmental drinking water samples (Table 5) (ANSES referral 2013-SA-0081) [14,16,17,18,19,20]. Then, the final concentrations in the animal drinking water were then calculated considering the water consumption of humans (estimated at 2 L per day for a reference weight of 60 kg), of mice (6 mL of water/day for a 40 g mouse) [146], and the different pharmacokinetics of each species (10 times higher in mice than in humans [147]). Thus, the calculated doses in the animal drinking water were: (i) D1: IBU 11.3 ng/L/2hIBU 90 ng/L + DCF 22.5 ng/L + EE2 2.25 ng/L; and (ii) D2: IBU 113 ng/L/2hIBU 225 ng/L + DCF 112 ng/L + EE2 45 ng/L (Table 5).

IBU (CAS 15687-27-1, >98% purity, I4883), 2h-IBU (CAS 51146-55-5, 32451), DCF sodium salt (CAS 15307-79-6, >98% purity, D6899) and EE2 (CAS 57-63-6, >98% purity, E4876) were purchased from Sigma-Aldrich (St. Louis, MO, USA). Stock solutions of IBU (5 mg/mL), 2h-IBU (2 mg/mL), DCF (2 mg/mL), and EE2 (1 mg/mL) were prepared in ethanol and kept at −20 °C in brown-colored bottles. Diluted solutions (11.3, 90, 22.5, 11.2, and 4.5 μg/mL or 113, 225, 112, and 45 μg/mL, respectively) of each chemical were mixed to prepare the D1 and D2 solutions (each working solution contained the same volume of drug and vehicle solution: 1 μL of each diluted solution per liter of drinking water) to fill the PPSU bottles. The control group was exposed to diluted ethanol (0.001%). The drug cocktails/diluted ethanol were added to the drinking water of pregnant mice from 8.5 dpc and of the pups until sacrifice (30 dpp for females; 35 dpp for males).

Then, to study the effects of paternal and maternal exposure, F1 males (m) or F1 females (f) that were exposed to the D1 or D2 doses (mD1, mD2, fD1 and fD2) were mated with male (mC) or female (fC) controls or with D2 animals (mD2 or fD2). In summary, these six mating combinations (i.e., mCxfC, mCxfD1, mCxfD2, mD1xfC, mD2xfC, and mD2xfD2) were used to generate F2 animals. To lighten the text, these mating combinations were indicated as CxC, CxD1, CxD2, D1xC, D2xC, and D2xD2, respectively (Appendix A). All in all, 21–46 females and 28–53 males were obtained for each exposure group for the F1 (7 L per group) and F2 (5 L per group) generations.

### 4.2. Tissue Collection, Oocytes Isolation and Histopathological Analysis

For each litter (F1 and F2), at least two 30 dpp females and two 35 dpp males (*n* >12 animals per group) were sampled. Animals and gonads were weighed and the AGD was measured. The genital tract and gonads were collected: testes and epididymis in males and ovaries in females. One gonad was fixed in 4% paraformaldehyde (in PBS) for histological analysis and the second was immediately frozen in dry ice and conserved at −80 °C for RNA extraction or hormone assays. Oocytes from F1 30 dpp females not stimulated with gonadotropins were isolated (*n* = 2–4 animals for each group) under a binocular magnifying glass via mechanical dissociation of the ovaries in M2 medium and frozen at −80 °C for RNA isolation. At this stage, follicles at the antral stage contain oocytes (germinal vesicle oocytes) blocked in prophase I of meiosis in which transcription is inactive.

Paraffin sections (3 μm thick) of F1 and F2 gonads were stained with hematoxylin-eosin using standard protocols to evaluate the tissue histology (RHEM Histology platform, Montpellier). Stained slides were scanned with a Nanozoomer Hamamatsu device (Hamamatsu Photonics, Tokyo, Japan) and analyzed with the Nanozoomer Digital Pathology software (NDPview2) (Hamamatsu Photonics). For IF analysis, tissue sections were deparaffinized, followed by antigen retrieval with 10 mM citrate buffer (pH 6) in the Histo5 apparatus (Milestone, Brondby, Denmark) for 20 min, as previously described [127,148]. Sections were then blocked with 10% donkey serum (Sigma-Aldrich) or 1/1000 anti-goat IgG (Thermofisher, Waltham, MA, USA), and incubated (4 °C, overnight) with the following primary antibodies: monoclonal mouse anti-PCNA (Sigma-Aldrich P8825, 1/500), anti-phospho-gH2Ax (Ser139 clone JBW301, Sigma Aldrich 05-636, 1/500), anti-COUPTFII (RD systems PP-H7147-00, 1/400), a-tubulin (Sigma Aldrich T9026, 1/1000), monoclonal rat anti-phosphorylated-Histone H3 (Ser10, Sigma Aldrich H9908, 1/400), anti-Tra98 (Abcam ab82527, 1/400), polyclonal guinea pig anti-PADI6 (1/500), polyclonal goat anti-Sox9 (RD systems AF3075, 1/300), polyclonal rabbit anti-FoxL2 (homemade [149], 1/300), anti-AMH (gift from N DiClemente, 1/300), anti-Lin28a (ProteinTech 11724, 1/400), anti-connexin-43 (Sigma Aldrich C6219, 1/500), and anti-a Smooth muscle actin (SMA) (Abcam ab124964, 1/500). Sections were then incubated with donkey or goat secondary antibodies conjugated with Alexa Fluor 555, 488, or 649 (Thermofisher, 1/1000) at room temperature for 30 min, followed by Hoechst staining. Cell apoptosis was assessed with the Dead-End Fluorometric TUNEL Apoptosis Assay kit according to the supplier’s procedure (Promega, G3250). After mounting in a Fluorsave mounting medium, IF images were captured with a Zeiss AxioImager apotome microscope (Carl Zeiss Microscopy, Jena, Germany) at the IGH Imaging facility (BioCampus, Canterbury, UK). IF images were then processed with the OMERO software (OMERO web 5.5.1., University of Dundee and Open Microscopy Environment). Particularly, the numbers of round spermatids (testes) and of FoxL2^+^ granulosa cells (from antral follicles) were counted in the regions of interest (ROIs) of a constant surface (4 mm^2^ for testes and 5 mm^2^ for ovaries). These ROIs (*n* = 4–10 per section) were selected with the OMERO software, in two to three sections of each testis/ovary (*n* = 3–5 per group); the number of phosphorylated H3^+^ granulosa cells was counted in at least three antral follicles in two sections (*n* = 4 animals per group); the numbers of Lin28a^+^ and Sox9^+^ cells were counted in 4 to 6 (stages VI–VIII) tubules per sections (*n* = 2–3) from *n* = 4 F1 animals or *n* = 2–4 F2 animals; tubule diameters and number of abnormal tubules (germ cells in the lumen/total number of tubules) were determined in 4–6 animals/group, and the percentage of primordial follicles (PF)/total number of follicles was analyzed in 3 to 5 ovaries per group in hematoxylin-eosin-stained sections with the NDPview2 software. The analyzed sections were separated by at least 20 μm, and counting was performed by two experimenters blinded to the group. Then, all the data were analyzed with GraphPrism 9.

BTB integrity was assessed via the biotinylation of surface proteins in decapsulated testis with the EZ-Link^TM^ Sulfo-NHS-LC-Biotin reagent (sulfosuccinimidyl-6-(biotin-amido) hexanoate) (1 mg/mL in PBS pH 8 containing 1 mM CaCl_2_) (Thermo Scientific, 21335). After 30 min, the reaction was stopped with 100 mM glycine/PBS. Then, testes were fixed in Bouin’s fixative, embedded, and frozen into OCT media. Biotinylation was revealed via incubating frozen sections (10 μm) with a FITC 488-conjugated streptavidin (1/200) for 30 min and analysed using a Zeiss AxioImager apotome microscope. The level of protamine deposition on sperm chromatin, which is part of the process of the sperm chromatin packaging, was evaluated based on the accessibility of the Chromomycin A3 fluorochrome (CMA3) to bind to protamine-free sites in spermatid DNA [48]. Briefly, deparaffined and rehydrated tissue sections were incubated with 0.25 mg/mL CMA3/citrate phosphate buffer pH7 and 10 mM MgCl_2_, at room temperature for 20 min. After washing in the same buffer, slides were mounted in Fluorsave and analyzed using a Zeiss AxioImager apotome microscope.

### 4.3. Anogenital Distance Measurement

AGD was measured at21 dpp and 30 (females) and at 35 dpp (males) offspring with a digital caliper from the center of the anus to the posterior edge of the genital papilla. Meanwhile, body weights were obtained. AGD was divided by the body weight (AGD/body Mw) [150].

### 4.4. Hormone Concentration Quantification

Blood samples were collected from 30 dpp F1 and F2 females and from 35 dpp F1 and F2 males via cardiac punction in Microvette Blood collection tubes (Sarstedt, Nümbrecht, Germany). Samples were allowed to clot for 60 min and were then centrifugated at 2000 g for 20 min; serum was frozen at −80 °C until hormone dosage. The concentration of FSH and LH in serum (50 μL) was measured with specific EIA ELISA kits (ElabScience Biotech, E-EL-M0511 and E-EL-M3053, respectively, CliniSciences, Nanterre, France) according to the manufacturer’s instructions (*n* = 1 to 6 animals per group). The mean sensitivities of these assays are 0.96 ng/mL for mouse FSH and 0.28 ng/mL for mouse LH. 17β-estradiol (females and testosterone males) was quantified using ELISA kits (Cayman 501,890 and Cayman 582,701, respectively) according to the manufacturer’s instructions (*n* = 1 to 6 animals per group for testosterone and *n* = 3 to 5 animals per group for 17β-estradiol). The mean sensitivities of these assays are 20 pg/mL and 6 pg/mL for 17b-estradiol and testosterone, respectively.

### 4.5. RNA Extraction, RT-qPCR and Sequencing (RNA-seq)

The total RNA (mRNA and long noncoding RNA > 200 bp)) from oocytes (*n* = 17 to 44 pooled from the F1 control, D1 and D2 groups), F1 and F2 35 dpp testes, and 30 dpp ovaries (*n* = 3 pools of 3 to 5 testes/ovaries from different mice/litter) was extracted and purified using the TRIZOL technique (Invitrogen). RNA purity was confirmed using the A260/A280 and A260/A230 ratios of >1.8 and >2.0, respectively. RNA integrity of each sample was measured using an Agilent 2100 BioAnalyzer (Agilent, Santa Clara, CA, USA). Real-time RT-PCR of oocyte RNA was performed as previously described [127,148]. For PCR, *18S* was used as the reference gene for data normalization; primer sequences were as follows: *Padi6*, forward, 5′-aaggggccacagggttccat and reverse, 5′-cgtggcaatcctcagctcca; *Zp1*, forward, 5′-aaggggccacagggttccat and reverse, 5′-cgtggcaatcctcagctcca; *Zp2*, forward, 5′-ggcccactggtgttggtcct and reverse, 5′-cagaggccgggtcctcagaa; *Nlrp14*, forward, 5′-ggcatgggctacgagtgcaa and reverse, 5′-gcccacaggtatgcgtggaa.

For RNA-seq experiments, libraries were constructed using the Stranded mRNA Prep Ligation kit (Illumina, San Diego, CA, USA) according to the manufacturer’s instructions. Briefly, poly-A RNA was purified using oligo-d(T) magnetic beads from 700 ng of total RNA. Poly-A RNA was fragmented and reverse-transcribed using random hexamers. During the second strand generation step, dUTP-substituted dTTP prevented the second strand being used as a matrix during the final PCR amplification. Double-stranded cDNA was adenylated at their 3′ ends and ligated to Illumina’s universal anchors. Ligated cDNA samples were amplified in 12 PCR cycles. During this step, the libraries were indexed, and the adapters sequences were completed in order to be compatible with the cluster generation step. PCR products were purified using AMPure XP Beads (Beckman Coulter Genomics, Brea, CA, USA). Libraries were validated using the Standard Sensitivity NGS kit on a Fragment Analyzer (Agilent Technologies, Santa Clara, CA, USA) and quantified using the KAPA Library quantification kit (Roche, Basel, Switzerland, CHE).

For sequencing, 54 libraries were pooled in equimolar amounts. The balance between all samples of the pool was assessed via sequencing on a Miniseq (Illumina, San Diego, CA, USA) using a 300 cycles Mid Output Reagent Cartridge (300 cycles). The pool was then sequenced on a Novaseq 6000 (Illumina, San Diego, CA, USA) on one lane of a S4 flow cell in the paired-end 2 × 100 nt mode according to the manufacturer’s instructions. This sequencing produced between 43 and 163 million passed filter clusters per library. For sequencing quality control, image analyses and base calling were performed using the NovaSeq Control Software and Real-Time Analysis component (Illumina, San Diego, CA, USA). Demultiplexing and trimming were performed using the Illumina’s conversion software (bcl2fastq 2.20). The raw data quality was assessed using FastQC (version 1.19) from the Babraham Institute and the Illumina software SAV (Sequencing Analysis Viewer). FastqScreen (version 0.15) was used to estimate the potential level of contamination.

For sequences alignment, the splice junction mapper, TopHat 2.1.1 [151], and Bowtie 2.4.5 [152]) were used to align the RNA-Seq reads to the *Mus musculus* genome (NCBI mm39) with a set of gene model annotations (gtf file up to date on 3 June 2022). Final read alignments with more than six mismatches were discarded. Samtools (v1.13) was used to sort the alignment files. Then, genes were counted with Featurecounts 2.0.3 [153]. As data were from a strand-specific assay, reads were mapped to the opposite strand of the gene (-s 2 option). Before statistical analysis, genes with <15 reads (cumulating all the analyzed samples) were filtered out. Differentially expressed genes were identified using the Bioconductor [154], package DESeq2 1.32.0 [155], and edgeR 3.34.1 [156] (R version 4.1.1). Data were normalized using the DESeq2 normalization method. Genes with adjusted *p*-values below 5% (according to the FDR method from Benjamini-Hochberg) were considered differentially expressed.

### 4.6. Statistical Analyses

Statistical analyses were performed with GraphPad Prism 7 software. One-way ANOVA and the Tukey’s post hoc test were used for multiple comparisons. All values are represented as the mean ± SEM of several independent experiments (*n* > 3). *p* < 0.05 was considered significant.

## 5. Conclusions

In this study, we identified the AOP networks for NSAIDs and EE2 at environmentally relevant doses in mice (Figure 11). Exposure to these chemicals modified the gonad transcriptome, including the expression of genes associated with human pathologies. The observed modifications of the testis and ovary physiology are sufficient to disturb puberty onset, which is a sensor of gene interaction perturbation during early development [157], in F1 and also in F2 females. The next step is to elucidate the mechanisms of these intergenerational effects. This study will also improve the AOP network of the human reproductive system development, particularly concerning EDCs, and may serve to identify putative EDCs for mammalian species using the gene expression of biomarkers.

## Figures and Tables

**Figure 1 ijms-24-05890-f001:**
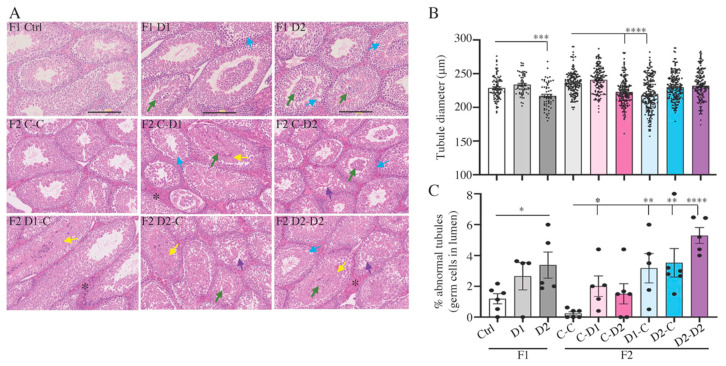
Histological analysis of 35 dpp F1 and F2 testes. (**A**) Representative images of hematoxylin-eosin-stained paraffin-embedded testis tissue sections. Scale bars: 200 μm. Green arrows: abnormal germ cell localization in the lumen; blue arrows: abnormal separation of germ cells from the basal compartment; purple arrows: vacuolization; yellow arrows: secretion; black stars: hyperplasic interstitial compartment. (**B**) Tubule diameter in the different conditions, *n* = 10–20 tubules/testis. Black dots are individual measurements. (**C**) Percentage of abnormal tubules containing germ cells in the lumen. (**B**,**C**) Each bar represents the mean ± SEM (*n* = 4–6 animals/group). ****: *p* < 0.001; ***: *p* < 0.005; **: *p* < 0.01; *: *p* < 0.05.

**Figure 2 ijms-24-05890-f002:**
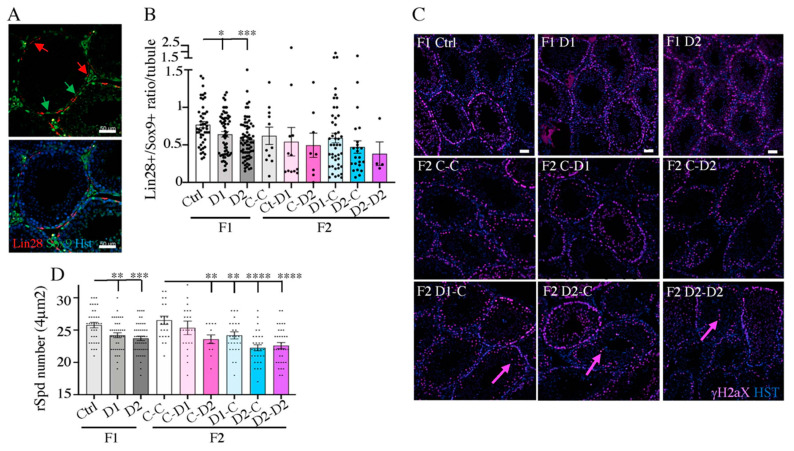
Analysis of spermatogenic cells via immunofluorescence staining of 35 dpp F1 and F2 testes. Paraffin-embedded testis tissue sections were incubated with antibodies against (**A**) Lin28a (undifferentiated spermatogonia marker; red) and SOX9 (Sertoli cell marker; green). Scale bars: 50 μm. (**B**) Ratio between the number of Lin28a^+^ cells and SOX9^+^ cells per tubule (*n* = 4–6 tubules/testis, *n* = 2–4 testis/group and n= 4 to 45 counted tubules). (**C**) γH2aX (double-strand break marker during meiosis) immunofluorescence analysis. Purple arrows: central positive cells (apoptotic cells). Scale bars: 100 μm. Nuclei were counterstained with Hoechst (HST; blue). ((**D**) Number of round spermatids (rSpd) determined by counting cells in 4–10 ROIs of 4 μm^2^ per testis (*n* = 3 to 5 testis/group). Data are the mean ± SEM. Black dots are individual measurements. ****: *p* < 0.001; ***: *p* < 0.005; **: *p* < 0.01; *: *p* < 0.05.

**Figure 3 ijms-24-05890-f003:**
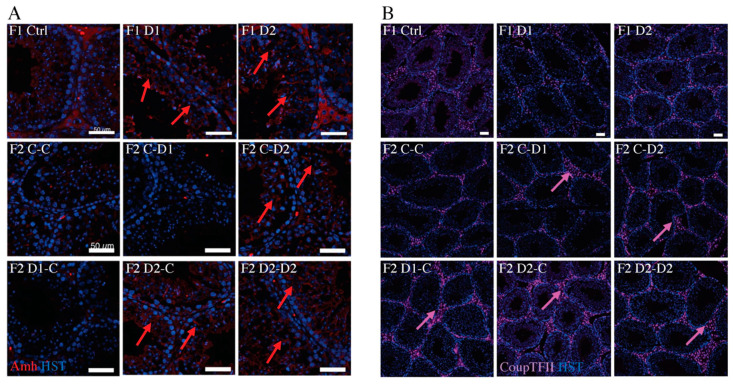
Immunofluorescence analysis of Sertoli and Leydig cells in 35 dpp F1 and F2 testes. Paraffin-embedded testis tissue sections were incubated with antibodies against (**A**) AMH (immature Sertoli cell marker; red). Scale bars: 50 μm, and (**B**) COUPTFII (Nr2f2; Leydig cell marker; purple). Pink arrows: central positive cells (apoptotic cells). Scale bars: 100 μm. Nuclei are counterstained with Hoechst (HST; blue).

**Figure 4 ijms-24-05890-f004:**
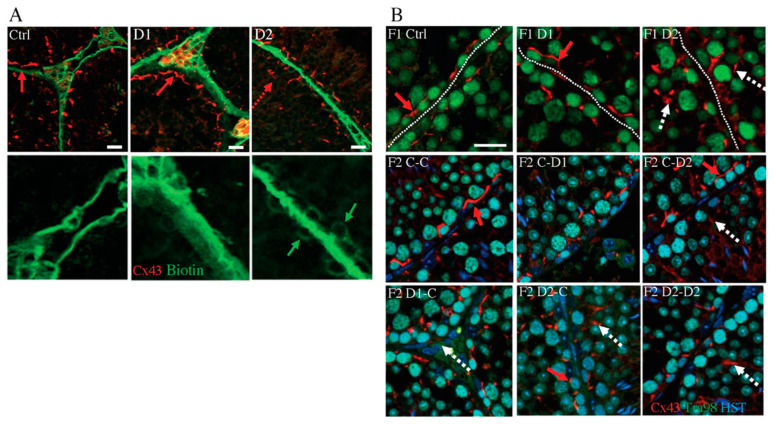
Analysis of the establishment and integrity of the blood–testis barrier in 35 dpp testes. (**A**) The biotin tracer was revealed with FITC-conjugated streptavidin (in green) and gap junctions were detected with an antibody against connexin 43 (Cx43; red) in F1 testes. Scale bars: 50 μm. (**B**) Immunostaining of Cx43 (gap junction localization) in F1 and F2 testes (red) and TRA98 (germ cells; green). Red arrows: normal basal Cx43 localization; dashed red arrows: abnormal Cx43 staining within tubules; dashed white arrows: abnormal Cx43 staining around germ cells. Scale bars: 20 μm. Nuclei were counterstained with Hoechst (HST; blue).

**Figure 5 ijms-24-05890-f005:**
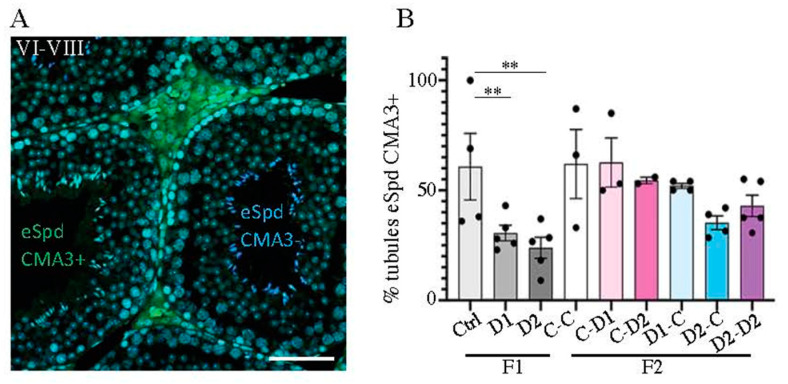
Effects on spermiogenesis onset in 35 dpp F1 and F2 testes. (**A**) Paraffin-embedded testis tissue sections were incubated with chromomycin A3 (CMA3) that fixes chromatin on protamine-free sites and CMA3 was revealed via fluorescence (l: 488); elongated spermatid (eSpd) in stages VI-VII tubules (eSpd stage 16) that were (or were not) labeled by CMA3 (eSpd CMA3^+^ or CMA3^-^) are highlighted in green or in blue, respectively. Nuclei were counterstained with Hoechst (HST; blue). Scale bars: 100 μm. (**B**) Percentage of CMA3^+^ stages VI-VII tubules; data are the mean ± SEM (*n* = 3 to 5 animals/group, and *n* = 3–6 tubules/testis). **: *p* < 0.01.

**Figure 6 ijms-24-05890-f006:**
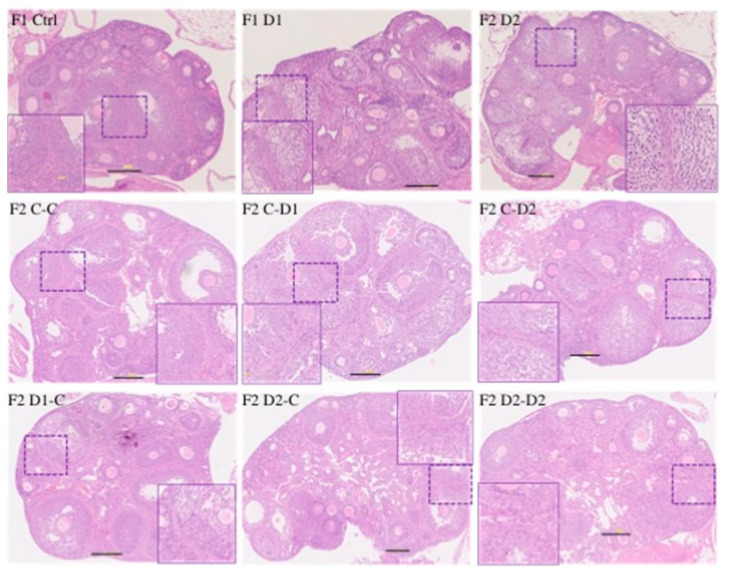
Histological analysis of 30 dpp F1 and F2 ovaries. Representative images of hematoxylin-eosin-stained paraffin-embedded ovary tissue sections. Scale bars: 200 μm. In each panel, the dashed box was enlarged (inset) to show follicular cells in secondary follicles.

**Figure 7 ijms-24-05890-f007:**
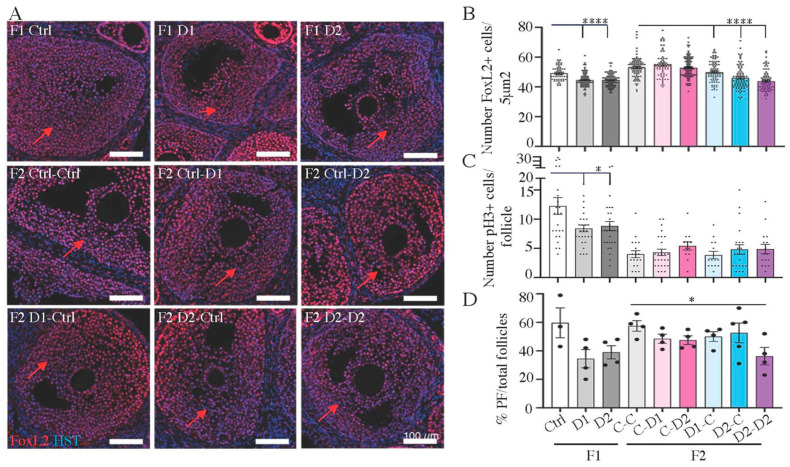
Folliculogenesis analysis in 30 dpp F1 and F2 ovaries via immunofluorescence staining. (**A**) Paraffin-embedded ovary tissue sections were incubated with an antibody against FOXL2 (granulosa cell marker; red); red arrows: granulosa cells in secondary follicles. Nuclei were counterstained with Hoechst (HST; blue). Scale bars: 100 μm. (**B**) Number of FOXL2^+^ granulosa cells in 4–8 ROIs of 5 μm^2^, *n* = 3–5 animals per group. (**C**) Proliferating cells in each secondary follicle were quantified using counting phospho-histone H3 (PH3)-positive cells (S-phase marker) (2 to 4 follicles per ovary, *n* = 3–5 animals/group) (Appendix A). (**D**) Percentage of primordial follicles (PF) relative to the total number of follicles in hematoxylin-eosin-stained ovaries (*n* = 3–5 animals/group) (Figure 6). (**B**–**D**) Data are presented as the means ± SEM. Black dots are individual measurements. ****: *p* < 0.001; ***: *p* < 0.005; **: *p* < 0.01; *: *p* < 0.05.

**Figure 8 ijms-24-05890-f008:**
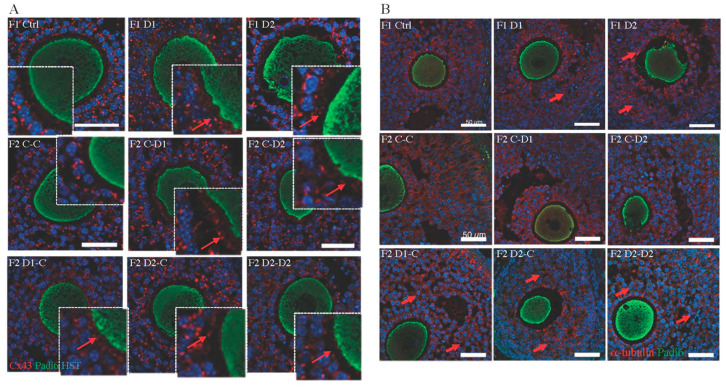
Organization of Cx43 (gap junction protein) and a-tubulin (cytoskeletal protein) in 30 dpp F1 and F2 ovaries. Paraffin-embedded ovary tissue sections were stained with antibodies against (**A**) Cx43 (red) and PADI6 (oocytes; green). Enlarged panels highlight the zona pellucida containing trans-zonal projections (TZPs); red arrows: increased Cx43 expression in F1 and F2 ovaries compared with controls. Scale bars: 20 μm; and (**B**) α-tubulin (red) and PADI6; red arrows: abnormal a-tubulin expression that delineates granulosa cell nuclei. Nuclei were counterstained with Hoechst (HST; blue). Scale bars: 50 μm.

**Figure 9 ijms-24-05890-f009:**
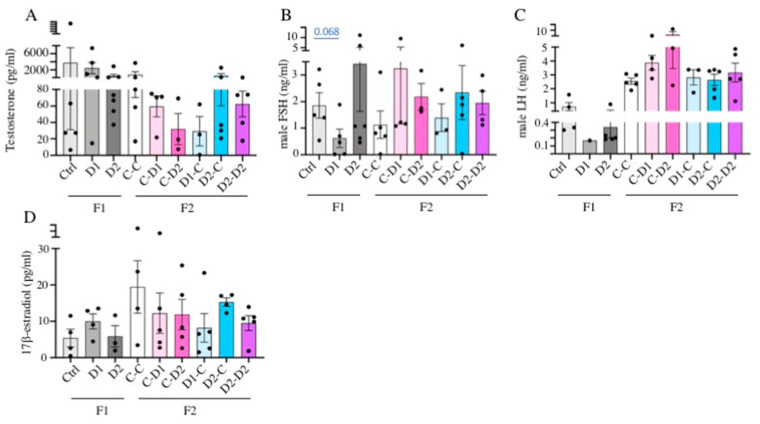
Hormone concentration in serum samples from F1 and F2 animals via ELISA. Serum (**A**) testosterone, (**B**) FSH, and (**C**) LH were measured in 35 dpp F1 and F2 males (*n* = 1 to 6 animals/group). (**D**) Serum 17β-estradiol was measured in 30 dpp F1 and F2 females (*n* = 3 to 5 animals/group). Data are the mean ± SEM. Black dots are individual measurements.

**Figure 10 ijms-24-05890-f010:**
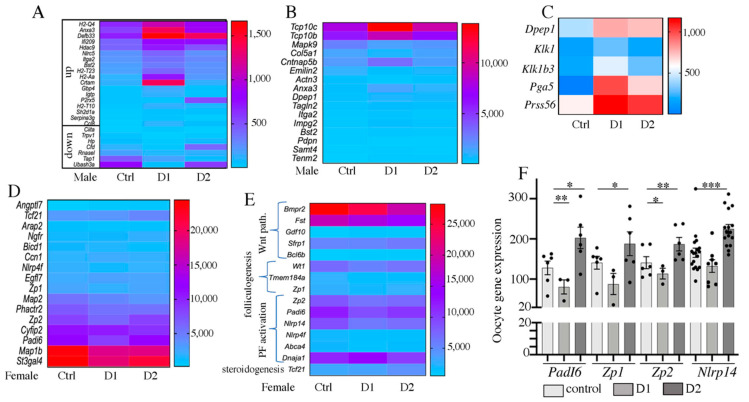
Heatmaps of DEGs in F1 males: (**A**) Immune and inflammatory genes; (**B**) cytoskeleton, microtubule, ECM genes; (**C**) proteases and peptidases genes. Heatmaps of DEGs in F1 females: (**D**) cytoskeleton, microtubule, ECM genes; (**E**) Wnt pathway, primordial follicle (PF) activation, folliculogenesis. (**F**) Oocyte gene expression in oocytes from 30 dpp F1 females via RT-qPCR; data were normalized to *18S* expression. Data are means ± SEM (*n* = 17 to 44 pooled oocytes for each group, *n* = 2 RT and at least 3 qPCR). ***: *p* < 0.005; **: *p* < 0.01; *: *p* < 0.05.

**Figure 11 ijms-24-05890-f011:**
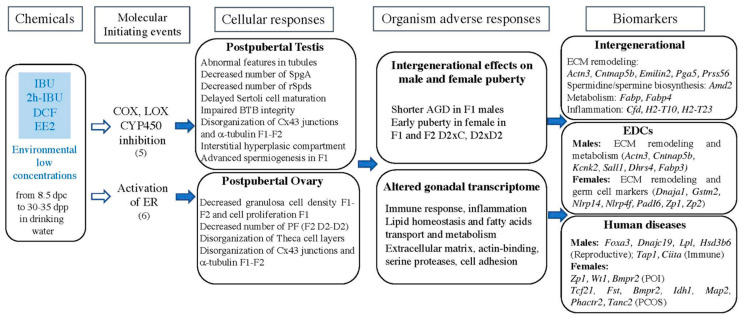
Adverse Outcome Pathway (AOP) for chronic exposure to NSAIDs and EE2 cocktails at environmentally relevant doses, and postnatal events that may influence adult health and diseases towards the identification of biomarkers of exposure. IBU: ibuprofen; 2h-IBU: 2 hydroxy-ibuprofen; DCF: diclofenac; EE2: ethinyl-estradiol; dpc: day post-coitum; dpp: day postpartum; COX: cyclooxygenases; LOX: lipoxygenase; CYP450: cytochrome P450; ER: estrogen receptors; PF: primordial follicle, SpgA: spermatogonia A; rSpd: round spermatid; BTB: blood–testis barrier; AGD: anogenital distance; Cx43: connexin 43; ECM: extra-cellular matrix; EDC: endocrine disrupting chemicals; PCOS: polycystic ovarian syndrome; POI: premature ovarian insufficiency.

**Table 1 ijms-24-05890-t001:** Descriptive statistics for male and female F1 animals: body weight (BW), gonad weight, anogenital distance (AGD), and AGD/BW ratio (^a^: *p* < 0.001; ^c^: *p* < 0.01; ^d^: *p* < 0.05). D1: exposure to dose D1, D2: exposure to dose D2. Bold numbers indicate significantly different values compared with controls.

F1 Animals	Males						Females					
	Control		D1		D2		Control		D1		D2	
	Mean	SEM	Mean	SEM	Mean	SEM	Mean	SEM	Mean	SEM	Mean	SEM
Body weight at 21 dpp (g) (*n*)	7.739 (51)	0.229	**9.557 ^a^** (30)	0.263	**8.865 ^c^** (37)	0.299	7.46 (40)	0.247	**8.693 ^c^** (30)	0.256	8.065 (43)	0.275
Body weight at 35/30 dpp (g) (*n*)	27.29 (11)	0.666	29.71 (8)	0.757	28.37 (10)	1.438	19.58 (5)	0.685	21.06 (5)	0.933	19.76 (5)	0.903
Gonad weight at 35/30 dpp (mg) (*n*)	79.91 (11)	4.308	86.21 (8)	3.982	85.85 (6)	6.009	2.96 (5)	0.326	2.3 (3)	0.321	1.94 (5)	0.188
Gonad weight/BW ratio (10^−3^) (*n*)	2.912 (11)	0.107	2.919 (8)	0.168	3.072 (6)	0.124	1.508 (5)	0.156	1.082 (3)	0.157	**0.981 ^d^** (5)	0.089
AGD/BW ratio at 21 dpp (10^−3^) (*n*)	1.017 (51)		**0.881 ^a^** (30)		**0.906 ^a^** (37)		0.565 (40)	0.013	0.541 (30)	0.013	0.552 (43)	0.012
AGD/BW ratio at 35 dpp (10^−3^) (*n*)	0.4802	0.011	0.472	0.015	0.466	0.014	-	-	-	-	-	-

**Table 2 ijms-24-05890-t002:** Descriptive statistics for male F2 animals, body weight (BW), gonad weight, anogenital distance (AGD), and ratio AGD/BW (^c^: *p* < 0.01; ^d^: *p* < 0.05). F1 exposed males (mD1 or mD2) and females (fD1 or fD2) were mated with male (mC) or female (fC) controls or with D2 animals (mD2 or fD2) to generate F2 animals: mCxfC (CxC), mCxfD1 (CxD1), mCxfD2 (CxD2), mD1xfC (D1xC), mD2xfC (D2xC), and mD2xfD2 (D2xD2). Bold numbers indicate significantly different values compared with controls.

F2 Males	CxC		CxD1		CxD2		D1xC		D2xC		D2xD2	
	Mean	SEM	Mean	SEM	Mean	SEM	Mean	SEM	Mean	SEM	Mean	SEM
Body weight at 21 dpp (g) (*n*)	12.27 (22)	0.524	12.41 (22)	0.376	11.97 (35)	0.357	**10.22 ^c^** (42)	0.368	11.27 (37)	0.30	12.31 (32)	0.439
AGD/BW ratio at 21 dpp (10^−3^) (*n*)	0.7509 (22)	0.02	0.7402 (28)	0.011	0.762 (35)	0.017	0.802 (41)	0.0178	0.797 (37)	0.014	0.758 (32)	0.016
Body weight at 30 dpp (g) (*n*)	30.93 (6)	1.248	29.20 (5)	0.671	27.5 (6)	1.018	25.96 (7)	2.189	28.20 (6)	1.213	29.93 (6)	0.908
Testis weight at 30 dpp (mg) (*n*)	95.60 (6)	4.84	92.06 (5)	4.362	85.43 (6)	3.927	**73.14 ^d^** (7)	7.608	83.83 (6)	5.799	83.92 (6)	2.289
Testis weight/BW ratio (10^−3^) (*n*)	3.083 (6)	0.04	3.162 (5)	0.177	3.113 (6)	0.121	2.807 (7)	0.198	2.957 (6)	0.093	2.810 (6)	0.077

**Table 3 ijms-24-05890-t003:** Descriptive statistics for female F2 animals, body weight (BW), gonad weight, anogenital distance (AGD), and AGD/BW ratio (^b^: *p* < 0.005; ^d^: *p* < 0.05). Bold numbers indicate significantly different values compared with controls.

F2 Females	CxC		CxD1		CxD2		D1xC		D2xC		D2xD2	
	Mean	SEM	Mean	SEM	Mean	SEM	Mean	SEM	Mean	SEM	Mean	SEM
Body Weight 21 dpp (g) (*n*)	12.93 (10)	0.847	11.39 (15)	0.511	**10.92 ^d^** (34)	0.352	**10.92 ^d^** (25)	0.357	**10.53 ^b^** (30)	0.306	11.48 (32)	0.45
AGD/BW ratio at 21 dpp (10^−3^) (*n*)	0.4438 (11)	0.024	0.4641 (21)	0.016	0.457 (34)	0.014	0.463 (25)	0.013	0.495 (30)	0.015	0.458 (32)	0.013
Body weight at 30 dpp (g) (*n*)	21.25 (4)	2.382	19.30 (5)	1.449	16.87 (7)	1.15	21 (6)	0.859	19.67 (6)	1.051	20.06 (7)	0.598
Ovary weight at 30 dpp (mg) (*n*)	1.525 (4)	0.154	1.38 (5)	0.162	1.3 (7)	0.198	1.883 (6)	0.122	1.48 (5)	0.156	1.617 (6)	0.282
Ovary weight/BW ratio (10^−3^) (*n*)	0.0769 (4)	0.016	0.0719 (5)	0.0071	0.0743 (7)	0.0078	0.0906 (6)	0.006	0.0079 (5)	0.009	0.0781 (6)	0.001

**Table 4 ijms-24-05890-t004:** Puberty onset (dpp) in F1 and F2 females, evaluated as the percentage (%) of vaginal opening (VO).

VO dpp	F1 Females			F2 Females					
%VO	Ctrl	D1	D2	CxC	CxD1	CxD2	D1xC	D2xC	D2xD2
onset	28	28	25–26	28	28	28	28	27	26
40%	35	34	32	33	33	34	34	34	31
60%	36	35	33	34	35	36	36	35	32
80%	38	36	35	36	36	38	37	37	33
90%	39	37	36	38	38	38	38	37	34

**Table 5 ijms-24-05890-t005:** Composition of the cocktail (D1 and D2) based on environmental human exposure data. Underlined numbers indicate the selected concentration used in the D2 dose mixture.

	CAS Number	Concentrations in Drinking Water (ng/L)		Human Exposure (ng/kg/Day)		Mouse Exposure (ng/kg/Day)		Exposure Concentration in Mice (ng/L)	
		Average	Maximum	Average	Maximum	Average	Maximum	Dose1 (D1)	Dose2 (D2)
IBU	15687-27-1	5	50–100	0.17	1.7	0.068	0.68	11.3	113
2h-IBU	51146-55-5	40	85–100	1.36	3.4	0.544	1.2	90	225
DCF	15307-86-5	10	50	0.34	1.7	0.136	0.68	22.5	112
EE2	57-63-6	1–2	20–50	0.068	0.68	0.0136	0.27	2.25	45

## Data Availability

The RNA-seq data have been deposited in the Gene Expression Omnibus database at NCBI and the accession number will be provided as soon as it becomes available.

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
