# Peer review of "Cocktails of NSAIDs and 17α Ethinylestradiol at Environmentally Relevant Doses in Drinking Water Alter Puberty Onset in Mice Intergenerationally"

_ijms, 2023, doi:10.3390/ijms24065890_

Round 1

Reviewer 1 Report

This paper assessed the effects of long-term exposure to the IBU, DCF, EE2 and 2h-IBU mixture on the reproductive tract of in utero exposed F1 mice. Overall, the paper is well written. 

Some specific comments are below:

1. Can authors explain more what does 

CxC CxD1 CxD2 D1xC D2xC D2xD2

mean in table 2 and 3, perhaps in a table footnote?

2. In Table 5, what does the underline mean?

3. In the statistical analysis part, the author first stated “Tukey's post-hoc test were used for multiple comparisons”, then later said “adjusted P-value (Bonferroni correction)“, which seems contradictory to each other. Please be clear which correction method was used.

Author Response

This paper assessed the effects of long-term exposure to the IBU, DCF, EE2 and 2h-IBU mixture on the reproductive tract of in utero exposed F1 mice. Overall, the paper is well written.

Some specific comments are below:

  1. Can authors explain more what does CxC CxD1 CxD2 D1xC D2xC D2xD2 mean in table 2 and 3, perhaps in a table footnote?

Response 1: We apologize for not having described the F1 and F2 mating combinations. 

“F1 males (m) or F1 females (f) that were exposed to the D1 or D2 doses (mD1, mD2, fD1 and fD2), were mated with males (mC) or females (fC) controls or with D2 animals (mD2 or fD2). In summary, these six mating combinations (i.e. mCxfC, mCxfD1, mCxfD2, mD1xfC, mD2xfC, and mD2xfD2) were used to generate F2 animals. To lighten the text, these mating combinations were indicated as CxC, CxD1, CxD2, D1xC, D2xC, and D2xD2, respectively”. This description is now introduced in the Materials and Methods section and in the legend of Table 2.

  1. In Table 5, what does the underline mean?

Response 2: We thank the reviewer to highlight this mistake, as we did not mention the significance of underlined numbers. We now modify the legend of Table 5: “Underlined numbers indicate the selected concentration used in the D2 dose mixture”.

  1. In the statistical analysis part, the author first stated “Tukey's post-hoc test were used for multiple comparisons”, then later said “adjusted P-value (Bonferroni correction)“, which seems contradictory to each other. Please be clear which correction method was used.

Response 3: We apologize for this mistake; after verification of the data analyzed in Graph Prism, the one-way ANOVA test with the Tukey’s post hoc test for multiple comparisons was used for all the data (cell counting, measurements of tubule diameters and RT-qPCR). This point is now modified in the revised manuscript.

Reviewer 2 Report

Author investigated that the effects of NSAIDs and EE2 chronic exposure in rodenet models. Especailly, on reproductive tract in F1 exposed mice and their F2 offspring. This manuscripts is quite interesting to me. I have a few question as follows: 

1) Figure 2. authour described figure legend (A)--> (C), please change this order A-B-C

2)Could you explain why reprsents ration between the number of Lin 28a cell and Sox9 cells per tubues

3) I think author can immunostaining with Lin28a (undifferentated germ cells), SCP3 or rH2aX (differentated germ cell), and Sox9 (Sertoli cell) and counting per tubules 

4)If you still have a samples could you show the gene expression levels of each marker from these testes?

5) In figure3. I think Amh antibody does not well working.. could you show the gene expression data additionally (A and B both)

6)Could you described that how did you counting the cell number from immunostaining image 

(slide number, area etc....)

7) In figure 10, how many samples (each groups) used for DEGs analysis. 

Author Response

Author investigated that the effects of NSAIDs and EE2 chronic exposure in rodenet models. Especailly, on reproductive tract in F1 exposed mice and their F2 offspring. This manuscripts is quite interesting to me. I have a few question as follows:

1) Figure 2. authour described figure legend (A)--> (C), please change this order A-B-C

Response 1: We thank the reviewer to highlight this mistake; we now modify the legend of Figure 2.  

2)Could you explain why reprsents ration between the number of Lin 28a cell and Sox9 cells perTubues

Response 2: Lin28 is expressed in undifferentiated and differentiating spermatogonia and is involved in undifferentiated spermatogonia proliferation (Chakraborty P 2014); thus, Lin28a is measuring the germ cell pool. Sertoli cells (Sox9+ cells) have functions in the process of testis formation in embryo and also in the spermatogenesis process. The number of Sertoli cells in postnatal testis determines the number of germ cells that can be supported through spermatogenesis in adult life. Indeed, each Sertoli cell has a fixed capacity for the number of germ cells that it can support (Orth JM 1998; Sharpe RM 1999).

Thus, the Lin28a+/Sox9+ cell ratio reflects the capacity of Sertoli cells to support spermatogenesis.

3) I think author can immunostaining with Lin28a (undifferentated germ cells), SCP3 or rH2aX (differentated germ cell), and Sox9 (Sertoli cell) and counting per tubules

Response 3: At 35 dpp, the first wave of spermatogenesis has started; however, all the tubules are not completely synchronized and on sections, the different stages of seminiferous epithelium cycle represented by different types of germ cells, are already identified. Thus, to evaluate the capacity of Sertoli cells to support germ cell differentiation all along the adult life, we used the ratio of Spermatogonia to Sertoli cells measured by the number of Lin28a+/Sox9+cells per tubule. This ratio is essential for an optimal sperm production and will determine the testis size.

We agree that counting of SCP3 and gH2aX would be also interesting since both are markers of spermatocytes during prophase I of meiosis; SCP3 being a major component of the synaptonemal complexes formed between homologous chromosomes, and phosphorylation of histone variant H2aX being characteristic to double-strand breaks. Instead, we have counted the number of round spermatids in Stage VI-VII tubules; round spermatids are the first haploid germ cells produced at the end of the meiosis process, and their number faithfully reflects the course of meiosis.

4)If you still have a samples could you show the gene expression levels of each marker from these testes?

Response 4: We agree with the reviewer comments, we could have analyzed the expression level of testicular markers. Instead of RT-qPCR, we choose to perform high-throughput sequencing which is more specific. These RNA-seq data are part of this manuscript.

Furthermore, we have done immunostaining analysis; these experiments allow to visualize the cellular localization of proteins while the expression of the genes is carried out on the whole tissues and does not inform on the posttranscriptional modifications (like the phosphorylation of H2aX).

However, we now introduced a HeatMap summarizing the expression data of some testicular markers that were obtained through RNA sequencing, in the revised manuscript (Supplementary Figure S2C); even these genes were not significantly differentially expressed, Amh and Nr2f2 (CouptfII) expression tended to increase and Lin28a and Sox9 expression tented to decrease, in F1-D1 and F1-D2 testes compared with controls.

5) In figure3. I think Amh antibody does not well working.. could you show the gene expression data additionally (A and B both)

Response 5: We agree with this comment; the AMH immunofluorescence labeling is weak, but it is specific since it is never visible in control testes. Furthermore, the AMH antibody provided by Dr N. di Clemente is highly specific and was successfully used in our previous experiments (Rossitto et al, Faseb J 2019).

Amh expression levels from high-throughput sequencing data shown on Supplementary Figure S2C, tend to increase in F1-D2 testis; however, these differences are not significant.

6)Could you described that how did you counting the cell number from immunostaining image (slide number, area etc....)

Response 6: As suggested, we now precise how the cells were counted from immunostaining images; in the revised manuscript, the paragraph 4.2 concerning cell counting was thus developed.

In summary:

“IF images were then processed with the OMERO software (OMERO web 5.5.1., University of Dundee and Open Microscopy Environment). Particularly, the numbers of round spermatids (testes) and of FoxL2+ granulosa cells (from antral follicles) were counted in regions of interest (ROIs) of a constant surface (4 mm2 for testes and 5 mm2 for ovaries). These ROIs (n=4-10 per section) were selected with the OMERO software, in two to three sections of each testis/ovary (n=3-5 per group); the number of phosphorylated H3+ granulosa cells was counted in at least three antral follicles in two sections (n=4 animals per group); the numbers of Lin28a+ and Sox9+ cells were counted in 4 to 6 (stage VI-VIII) tubules per sections (n=2-3) from n= 4 F1 animals or n=2-4 F2 animals; tubule diameters and number of abnormal tubules (germ cells in the lumen/total number of tubules) were determined in 4-6 animals/group, and the percentage of primordial follicles (PF)/total number of follicles was analyzed in 3 to 5 ovaries/group in hematoxylin-eosin stained sections with the NDPview2 software. The analyzed sections were separated by at least by 20 mm, and counting was performed by two experimenters blinded to the group. Then, all the data were analyzed with GraphPrism”.

7) In figure 10, how many samples (each groups) used for DEGs analysis.

Response 7: All RNA-seq experiments were performed in triplicates, since 3 RNA pools for each group were used; these pools were constituted by individually prepared RNAs from 3 to 5 testes/ovaries, from different mice/litter. DEGs analysis was thus performed between the 3 RNA replicates.

These technical points are described paragraph 4.5.

Reviewer 3 Report

In the current work presented by Philibert et al., showed that at chronic exposure of lower environmentally relevant doses, the cocktails of endocrine disrupting pharmaceuticals,17α-ethinyl-estradiol (EE2) and  non-steroidal anti-inflammatory drugs (NSAIDs) in drinking water alter puberty onset in mice. In males, it delayed puberty whereas in females, it accelerated puberty. Interestingly,  the modifications of testes and ovaries, differentiation/maturation of the different gonad cell types, were observed in the non-exposed F2 generation. Through transcriptomic analysis of post pubertal testes and ovaries of F1 (exposed) and F2 animals, the authors found that significant changes occurred in gene expression profiles and enriched pathways, particularly the inflammasome, metabolism and extracellular matrix pathways, compared with controls (non-exposed).

The work is relevant and I have no suggestion before publication.

Author Response

In the current work presented by Philibert et al., showed that at chronic exposure of lower environmentally relevant doses, the cocktails of endocrine disrupting pharmaceuticals,17α-ethinyl-estradiol (EE2) and non-steroidal anti-inflammatory drugs (NSAIDs) in drinking water alter puberty onset in mice. In males, it delayed puberty whereas in females, it accelerated puberty. Interestingly, the modifications of testes and ovaries, differentiation/maturation of the different gonad cell types, were observed in the non-exposed F2 generation. Through transcriptomic analysis of post pubertal testes and ovaries of F1 (exposed) and F2 animals, the authors found that significant changes occurred in gene expression profiles and enriched pathways, particularly the inflammasome, metabolism and extracellular matrix pathways, compared with controls (non-exposed).

The work is relevant and I have no suggestion before publication."

Comment 1: The authors thank the reviewer for these comments.

Reviewer 4 Report

I have now completed reviewing the article “Cocktails of NSAIDs and 17a ethinyl-estradiol at environmentally relevant doses in drinking water alter puberty onset in mice inter-generationally”. I just pointed out several suggestions which may improve the quality of the manuscript. There were no major concerns in the manuscript. The topic of studying is interesting, and the data are robust. However, the reviewer thinks that the manuscript is too prolix result in reducing readability. If possible, you can reduce words again.

Author Response

I have now completed reviewing the article “Cocktails of NSAIDs and 17a ethinyl-estradiol at environmentally relevant doses in drinking water alter puberty onset in mice inter-generationally”. I just pointed out several suggestions which may improve the quality of the manuscript. There were no major concerns in the manuscript. The topic of studying is interesting, and the data are robust.

However, the reviewer thinks that the manuscript is too prolix result in reducing readability. If possible, you can reduce words again.

Response 1: We thank the reviewer for these comments; as suggested, we tried to reduce the text, particularly, in the Results section.

Reviewer 5 Report

Generally the MS is very interesting with many new original results.

There are some suggestions:

ABSTRACT

- is this really only related to aquatic ecosystem? (L29)

- "exposure to therapeutic doses has a negative impact" - is this really therapy (L30)

- explain for the first time - dpc, dpp

- give clear conclusion and novelty of this study (as well prepare in the final part of discussion)

INTRODUCTION

- give clearly aims of this study and possible hypothesis

RESULTS

- please give clearly: "presence of germ cells in the lumen of ST was abnormally increased..."

- L175: is it detachment of germ cells from the Sertoli epithelium? - or detachment of germinal epithelium from the basal membrane

- Figure 1: give clearly "germ cells in the tubule lumen"

- as there are also citation in the section Results, a connection of chapters to Results and Discussion should be clearer

DISCUSSION

- the last part (L665-672) should be a separate chapted Conclusions

MATERIAL AND METHODS

- check solutions concentrations (L707) - 11.5 and 4.5...

- some minor error (abbraviations give full for the first time)

Author Response

ABSTRACT

- is this really only related to aquatic ecosystem? (L29)

Response 1: We agree that this term is too exclusive. We focused on this point (drinking water) because our study mimics a chronic exposure to these molecules through drinking water.

This sentence has been changed: “NSAIDs and EE2 are among the most relevant endocrine disrupting pharmaceuticals found in the environment, particularly, in surface and drinking water”.

- "exposure to therapeutic doses has a negative impact" - is this really therapy (L30)

Response 2: We agree that this sentence is ambiguous; it has been changed to “Exposure of pregnant mice to NSAID therapeutic doses during the sex determination period, has a negative impact on gonadal development and fertility in adults”.

- explain for the first time - dpc, dpp

Response 3: As suggested, we now define the terms “dpc and dpp” when first appear in the text (days post-coitum and days post-partum, respectively). In the abstract, the term “dpc” was removed and the sentence was replaced by “added to the drinking water from fetal life until puberty”.

- give clear conclusion and novelty of this study (as well prepare in the final part of discussion)

Response 4: As suggested, on the end of abstract, we introduced a sentence that concludes this study and shows its novelty: “The identified Adverse Outcome Pathways (AOP) network for NSAIDs and EE2 at doses that are relevant to everyday human exposure, will improve the AOP network of the human reproductive system development, concerning endocrine disruptor chemicals. It may serve to identify other putative endocrine disruptors for mammalian species, based on the expression of biomarkers”.

INTRODUCTION

- give clearly aims of this study and possible hypothesis

Response 5: As suggested, we modified the last paragraph of the Introduction part, and highlighted the hypothesis of the study and its aims:

“On the basis of the published findings on NSAID and EE2 exposure, we hypothesized that long-term exposure to the IBU, DCF, EE2 and 2h-IBU mixture (at two environmentally relevant doses) could affect the reproductive organ development of in utero exposed F1 mice, leading to impaired puberty onset. Using cellular and transcriptomic approaches, we studied gonad development and gene expression in the exposed F1 generation and also, in the F2 generation (not directly exposed) to evaluate the potential inter-generational effects of these drugs. We identified the Adverse Outcome Pathway (AOP) network for NSAIDs and EE2 at environmentally relevant doses that may provide new insights for reproductive toxicity studies on other endocrine disruptors, based on biomarker transcriptomic analysis.”.

RESULTS:

- please give clearly: "presence of germ cells in the lumen of ST was abnormally increased..."

Response 6: This sentence was modified “presence of undifferentiated germ cells in the lumen was abnormally observed in F1-D2….”

- L175: is it detachment of germ cells from the Sertoli epithelium? - or detachment of germinal epithelium from the basal membrane

Response 7: Germ cells are detached from the Sertoli epithelium rather than the basal membrane. Some environmental chemicals are shown to impair the blood-testicular barrier (BTB) formation, leading to the loss of cell junctions. Indeed, our following experiments showed that BTB integrity was impaired in F1-D2 testes (Figure 4), and that gap junction (Cx43 staining, Figure 4B) and cytoskeleton organization (Tubulin staining, Supplementary Figure S3) may be affected by exposure.

This paragraph related to Figure 1 has been modified in the revised manuscript.

- Figure 1: give clearly "germ cells in the tubule lumen"

Response 8: This sentence is now modified to “abnormal germ cell localization in the lumen” in the legend of Figure 1. This abnormal feature was highlighted by green arrows.

- as there are also citation in the section Results, a connection of chapters to Results and Discussion should be clearer

Response 9: As suggested, the connection of both Results and Discussion parts has been revised and improved.

DISCUSSION

- the last part (L665-672) should be a separate chapted Conclusions

Response 10: As suggested, we introduced a separated Conclusions part.

MATERIAL AND METHODS

- check solutions concentrations (L707) - 11.5 and 4.5...

Response 11: We apologize because D1 concentrations were wrong: instead of 11.3, 90, 22.5, 11.2 and 4.5 mg/ml for IBU, 2hIBU, DCF and EE respectively, the concentrations were 11.3, 90, 22.5, 22.5 and 2.25 mg/ml, respectively, and according to Table 5.

- some minor error (abbreviations give full for the first time)

Response 12: As suggested, errors concerning abbreviations have been revised all along the manuscript.

Round 2

Reviewer 2 Report

congrats! Accept in present from